# RL with Learnable Textual Feedback: A Bilevel Approach

**Utsav Singh** [* 1]   **Sidhaarth Sredharan** [* 2]   **Souradip Chakraborty** [3]   **Amrit Singh Bedi** [1]

## Abstract

Reinforcement learning with verifiable rewards can improve LLM reasoning, but learning remains sample-inefficient when terminal rewards are sparse. This has motivated a growing line of work on RL with textual feedback, where a critic model generates natural language feedback to guide a reasoning model (the actor), augmenting scalar rewards with richer learning signals. However, existing methods typically treat feedback as fixed or auxiliary, which misses a key property: feedback should not merely be correct, but should improve the policy (actor model) when provided in context. This motivates a paradigm of *learnable textual feedback for RL*. Yet the learnability and usefulness of feedback depend on the policy's ability to learn from it, making RL with learnable feedback an inherently bilevel problem. We formalize this coupling as a Stackelberg bilevel program and derive *Bilevel Natural Language Actor-Critic* (Bi-NAC), which jointly trains a critic to generate reward-improving feedback and an actor to exploit it. Across MATH-500, MBPP, and GPQA, Bi-NAC improves sample and parameter efficiency over RL and fixed-critic baselines: our 2B model outperforms the 3B GRPO baseline, achieving 46.6% versus 41.4% on MATH-500, while our 6B model surpasses the 7B GRPO baseline, achieving 49.3% versus 43.6% on GPQA. Our code is available at https://github.com/sid-in-the-loop/Bi-Natural-Language-Actor-Critic/.

## 1. Introduction

Reinforcement learning with verifiable rewards has emerged as a powerful tool for improving large language model (LLM) reasoning, but it remains fundamentally limited by sparse, terminal supervision (Shao et al., 2024; Guo et al.,

[1]University of Central Florida [2]Carnegie Mellon University [3]Universtiy of Maryland. Correspondence to: Amrit Singh Bedi <amritbedi@ucf.edu>.

*Proceedings of the 43rd International Conference on Machine Learning*, Seoul, South Korea. PMLR 306, 2026. Copyright 2026 by the author(s).

2025). When the reward is binary and terminal, partially correct solutions receive no credit, and intermediate reasoning steps receive no direct feedback. In group-based objectives such as GRPO, this issue is amplified: if all sampled candidates for a prompt are incorrect, the group-normalized advantage collapses to zero, and the policy gradient update vanishes for that training step (Figure 1, Left).

**RL with textual feedback.** To densify supervision beyond terminal correctness, a growing line of work augments scalar rewards with natural language feedback, where a critic language model generates critiques that guide another language model (the actor) toward correct solutions. Such feedback has been explored in various forms, from predefined correction instructions (Lightman et al., 2024) to model-generated critiques (Zhang et al., 2026). However, existing methods typically treat feedback generation as a fixed or auxiliary component, with the critic model either frozen or trained with objectives decoupled from the actor's downstream performance. Recent work highlights this limitation from complementary angles: generative reward models can produce flawed critiques even when their outcome predictions are accurate (Wang et al., 2026b), while longer, unconstrained textual feedback can degrade policy performance (Lloret et al., 2024). Together, these findings suggest that the central challenge is not simply producing more feedback, or even correct feedback, but producing feedback that is actionable for the actor: a critique may be fluent, or locally correct, yet still fail to improve the actor's policy when provided in context.

**Our key insight: Learnable textual feedback requires coupled training.** We argue that feedback should be evaluated not by its standalone correctness, but by its downstream effect on the actor after adaptation. This shifts the design question from *how to produce correct feedback* to *how to produce feedback the actor can learn from*, motivating a paradigm of *learnable textual feedback for RL*. The learnability and usefulness of feedback are deeply entangled with the actor's learning dynamics: producing better feedback requires reasoning about how the actor will adapt to it, while the actor's adaptation depends on the feedback it receives. Resolving this coupling at the LLM scale is non-trivial, as it requires optimizing a critic LLM through the learning dynamics of an actor LLM, without sacrificing the simplicity that makes GRPO-style training practical.

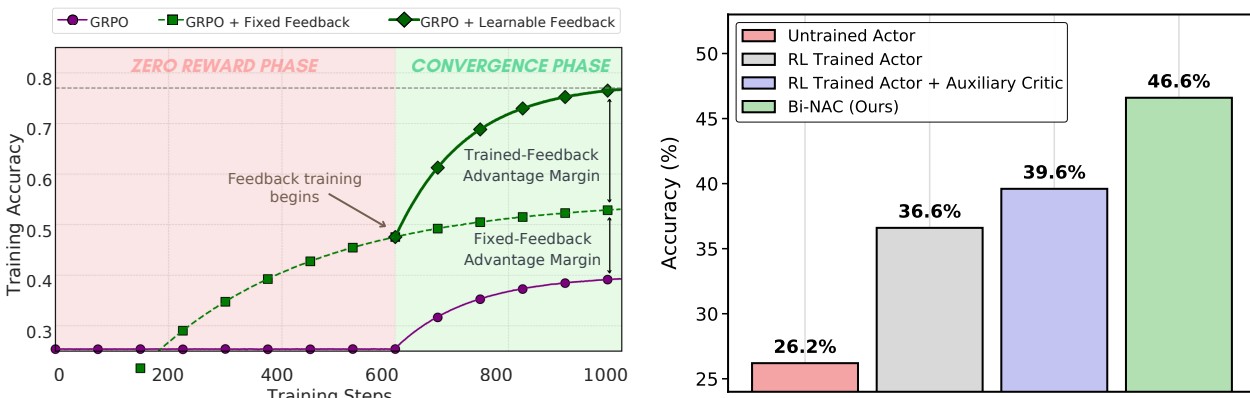

*Figure 1.* **Trained, aligned feedback yields faster convergence and higher reasoning accuracy.** (*Left*) Vanilla GRPO on MBPP with LLaMA-3.2-1B exhibits a *zero-reward phase* (∼600 steps) where advantage collapse stalls learning. *Fixed feedback* (critiques from a frozen LLM not optimized for task reward) yields only marginal gains (≈0.1), replacing it with learnable feedback, where the critic is optimized for the actor's downstream reward, yields a larger improvement (≈0.4). (*Right*) On MATH-500, an untrained actor reaches 26.2%, and vanilla RL reaches 36.6%; fixed feedback adds little (39.6%), but our Bi-NAC, which jointly trains actor and critic via a bilevel objective, reaches 46.6% (+7% over fixed feedback). **Takeaway:** feedback *quality* drives sample efficiency; aligning critic and actor through bilevel optimization leads to superior gains over static feedback-based methods.

**Proposed approach.** We address this entanglement by formulating *learnable textual feedback RL* as a Stackelberg bilevel program. At the lower level, the actor optimizes its policy to maximize task reward conditioned on the critic's feedback. At the upper level, the critic is optimized through the actor's learning dynamics, and is rewarded for producing feedback that improves the actor's downstream task reward after feedback-conditioned adaptation. This makes the dependency between critic feedback and policy learning explicit, directly targeting the failure mode of independently trained or fixed critics. We instantiate this view in *Bilevel Natural Language Actor-Critic (Bi-NAC)*, a practical algorithm for reasoning with verifiable rewards that jointly trains an actor LLM and a feedback-generating critic LLM end-to-end. We summarize our main contributions as follows.

- **A structural failure mode in RL with textual feedback.** We identify that independently trained critic LLMs can generate correct yet *ineffective* feedback that does not translate into improved actor performance, motivating the need for learnable feedback.

- **A bilevel framework for learnable textual feedback.** We propose Bilevel Natural Language Actor-Critic (Bi-NAC), a principled bilevel framework that explicitly models the coupling between feedback generation and policy learning. Bi-NAC optimizes the critic through the actor's downstream learning dynamics, training it to generate feedback that maximizes the actor's final task reward after adaptation.

- **Improved parameter and sample efficiency.** Across MATH-500, MBPP, and GPQA, Bi-NAC improves both parameter and sample efficiency over GRPO and

fixed-feedback baselines. Our 2B and 6B models outperform 3B and 7B GRPO baselines on the same benchmarks, highlighting the gains from *learnable feedback* aligned with the actor's learning dynamics.

## 2. Problem Formulation

We study reinforcement learning for LLM reasoning in settings with a verifiable reward. Given an input problem $x \sim \mathcal{D}$, a policy $\pi_\theta(\cdot \mid x)$ generates a response $y$ and receives a terminal reward $R(x,y) \in \{0,1\}$ that returns 1 if the final answer is correct and 0 otherwise. This binary terminal supervision is common in math and code tasks, but it creates a **sparsity bottleneck**. For each input $x$, GRPO samples a group $\{y_i\}_{i=1}^G \sim \pi_\theta(\cdot|x)$ and receives rewards $\mathcal{R}(x, y_i)$ returning 1 for correct solutions and 0 otherwise. The GRPO objective is given by

$$J_{\text{GRPO}}(\theta) = \mathbb{E}_{y \sim \pi_\theta(\cdot|x)}[R(x,y) - \hat{R}],$$

where $R(x,y)$ is the final task reward and $\hat{R}$ is a group-normalized baseline computed by $\hat{R} = \frac{1}{G}\sum_{i=1}^G R_i$. The group-normalized advantage for each sequence is computed as $A_i = \frac{R_i - \hat{R}}{\sigma_R + \epsilon}$, where $\sigma_R$ is the standard deviation of rewards within the group and $\epsilon$ is the numerical stability constant. GRPO's reliance on binary verification based sparse rewards creates a fundamental bottleneck: when all responses in a group are incorrect, the resulting zero advantages eliminate learning signals entirely. This advantage collapse occurs because GRPO computes relative rewards within each group: if every response receives $R_i = 0$, then reward $\hat{R} = 0$ and advantage $A_i = 0 \forall i$, providing no learning signal.

**Illustrative Example.** We present an example in Figure 1 (Left), where we train a GRPO-based policy using the LLaMA-3.2-1B model on a subset of the MBPP benchmark consisting of 200 reasoning examples. With a group size of $G = 4$ GRPO generates four candidate responses that all fail to solve the task. Although some of these responses are partially correct, each receives a sparse reward of $R_i = 0$ resulting in a collapsed advantage $A = 0$. To further substantiate this behavior, we conduct an empirical study on the same MBPP subset using the LLaMA-3.2-1B backbone, demonstrating the prevalence of reward and advantage collapse in GRPO, which in turn leads to slow learning and sample inefficiency. To quantify how often sparsity removes the learning signal in practice, we introduce two simple diagnostics computed during training on MBPP.

- **Advantage collapse frequency.** We empirically measure the *Advantage Collapse frequency*, which is the percentage of groups during training that lead to advantage $A = 0$. On the MBPP task, we observe the advantage collapse frequency to be $74\%$, which implies that $74\%$ of the training groups experience complete advantage collapse, leading to no learning signal for most training steps.

- **Reward collapse frequency.** We also measure the *Reward Collapse frequency*, which is the percentage of responses across training that lead to rewards $R = 0$. We observe that on MBPP task, the reward collapse frequency is $83\%$, implying that $83\%$ of the trajectories lead to 0 rewards, further confirming the prevalence of extremely sparse learning signals in GRPO.

This suggest a clear requirement: RL for reasoning requires denser learning signals rather than just terminal correctness. A natural approach is to introduce textual feedback that can guide the model toward improved second attempts, thereby increasing the probability of obtaining non-zero rewards and reducing the frequency of collapsed updates.

### 2.1. From Dense Textual Feedback to Learnable Feedback

We investigate the utility of leveraging textual feedback to improve performance on complex reasoning tasks. To this end, we consider a feedback-based setup: given an input $x$, the actor first produces an initial attempt $y_0$; a critic then analyzes this attempt to generate textual feedback $z$; finally, the actor conditions on this feedback to produce a refined response $y_1$. To quantify the utility of leveraging textual feedback, we analyze the impact of key feedback strategies on reasoning accuracy in Figure 1 (Right). Concretely, we compare four main approaches to isolate the benefits of feedback and bilevel optimization: (i) Untrained Actor, a pre-trained policy operating without feedback; (ii) RL-

Trained Actor, a policy trained using vanilla GRPO without feedback; (iii) RL Trained Actor + Auxiliary Critic, where the RL-trained actor utilizes critiques from a pre-trained LLM; and (v) Bi-NAC (Ours), which jointly optimizes both actor and critic via our bilevel formulation.

**Fixed feedback baseline.** We find that adding fixed or auxiliary feedback provides only limited improvement over GRPO. Although such feedback can partially densify the learning signal and speed up early training, its gains quickly saturate because the feedback is not optimized for the actor's downstream improvement. While dense feedback improves sample efficiency and speeds up convergence (Figure 1 Left), performance quickly saturates at around 0.5 training accuracy, compared to approximately 0.4 for GRPO, yielding only a modest gain of about 0.1. Test-time accuracy improvements are similarly marginal (Figure 1 Right), indicating that fixed auxiliary feedback is suboptimal and insufficient for meaningfully improving reasoning performance.

**Key challenge.** These results reveal a key limitation of textual feedback approaches that train the actor while keeping the critic fixed or only loosely coupled. Even when feedback is correct in isolation, it may be ineffective if the actor does not interpret it in a way that increases task reward. Effective learning with textual feedback requires two coupled conditions: (1) the critic must generate feedback that is not only plausible, but useful in the specific sense that conditioning on it improves the actor's downstream reward, and (2) the actor must be trained to exploit the feedback to increase the probability of correct verified outcomes. Because the critic's utility depends on how the actor adapts to its feedback, the two learning problems are inherently coupled, motivating a bilevel formulation for natural language actor-critic training.

## 3. Proposed Approach: Bi-NAC

We introduce **Bilevel Natural Language Actor-Critic (Bi-NAC)**, a principled end-to-end framework for learning with textual feedback in RL for LLM reasoning (Figure 2). Bi-NAC jointly trains (i) an actor LLM that generates and refines solutions and (ii) a critic LLM that produces feedback. The key design choice is that the critic is not trained to generate plausible feedback in isolation, but to generate feedback that improves the actor's verified task reward after the actor adapts to the feedback. Given an input prompt $x$, the actor $\pi_\theta^L$ produces an initial attempt $y_0 \sim \pi_\theta^L(\cdot \mid x)$. The critic $\pi_\phi^H$ generates feedback $z \sim \pi_\phi^H(\cdot \mid x, y_0)$. The actor then produces a refined response $y_1 \sim \pi_\theta^L(\cdot \mid x, y_0, z)$, which is evaluated by a verifiable reward $R(x, y_1) \in \{0, 1\}$.

**Why bilevel?** Training with fixed feedback treats $z$ as an external input, which can densify learning but often saturates (Figure 1, Right). Bi-NAC instead treats feedback as a

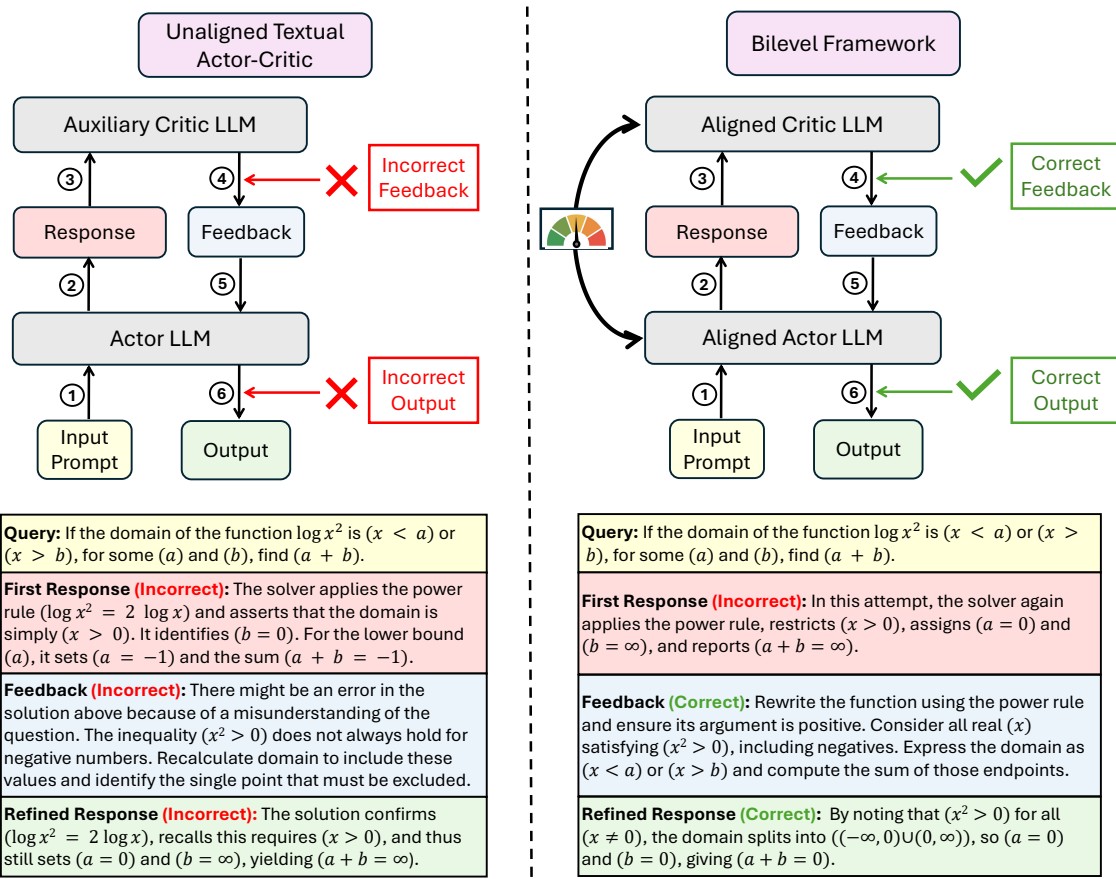

*Figure 2.* **Our Bilevel Framework.** Top Left: In prior approaches where actor and critic are not aligned (e.g., auxiliary critics or fixed feedback critics), this misalignment often leads to ineffective guidance, where the feedback is either incorrect or ignored by the actor, thus resulting in incorrect final outputs. Bottom Left: We show this in an illustrative example where the feedback and final output are incorrect. Top Right: Bi-NAC explicitly models the dependency between feedback generation and policy improvement by training the critic to maximize the final task performance of the actor, thus also enabling the actor to effectively leverage it. Bottom Right: The illustrative example shows how correct aligned feedback successfully guides the actor from a flawed initial response to the correct final solution.

learnable action that should be optimized for its effect on the actor's learning dynamics. This induces a natural leader-follower structure: the critic chooses feedback anticipating how the actor will learn from it, while the actor learns to exploit feedback to maximize verified reward.

### 3.1. Bi-Level Formulation of Natural Language Actor-Critic

**Lower-level actor LLM.** The actor optimizes its response generation to maximize the task reward, conditioned on the feedback generated by the upper-level critic. Concretely, for a fixed critic policy, the actor updates its parameters by maximizing the expected reward of the refined response:

$$\max_{\theta} L(\phi, \theta) := \mathbb{E}_{\substack{x \sim \mathcal{D}, \, y_0 \sim \pi_\theta^L(\cdot|x) \\ z \sim \pi_\phi^H(\cdot|x,y_0) \\ y_1 \sim \pi_\theta^L(\cdot|x,y_0,z)}} [R(x, y_1)] . \quad (1)$$

**Higher-level critic LLM.** The critic's goal is to maximize the final task performance achieved by the actor. Unlike standard RL critics that strictly minimize value prediction error, our critic is itself a policy optimized to generate feedback that improves the actor's expected reward:

$$\max_{\phi} U(\phi, \theta) := \mathbb{E}_{\substack{x \sim \mathcal{D}, \, y_0 \sim \pi_\theta^L(\cdot|x) \\ z \sim \pi_\phi^H(\cdot|x,y_0) \\ y_1 \sim \pi_\theta^L(\cdot|x,y_0,z)}} [R(x, y_1)] . \quad (2)$$

We formulate the reasoning process as a bi-level optimization problem, where the critic (higher-level policy) and the actor (lower-level policy) optimize coupled objectives. Formally, we cast learning as a Stackelberg game over the dataset $\mathcal{D}$:

$$\max_{\phi} U(\phi, \theta^*(\phi)) \quad \text{s.t.} \quad \theta^*(\phi) = \arg\max_{\theta} L(\phi, \theta), \quad (3)$$

where $\theta^*(\phi)$ denotes the optimal actor parameters conditioned on the critic's feedback policy. Following recent ad-

vances in bi-level optimization (Liu et al., 2022), the above problem can be equivalently expressed as a constrained optimization problem:

$$\max_{\phi,\theta}; U(\phi,\theta) \quad \text{s.t.} \quad L(\phi,\theta) - L(\phi,\theta^*(\phi)) \geq 0, \quad (4)$$

where the constraint is satisfied if and only if $\theta = \theta^*(\phi)$. This formulation allows us to derive the corresponding Lagrangian:

$$\max_{\phi,\theta} \mathcal{L}(\phi,\theta,\lambda) = \max_{\phi,\theta} U(\phi,\theta) + \lambda\big(L(\phi,\theta) - L(\phi,\theta^*(\phi))\big), \quad (5)$$

where $\lambda \geq 0$. We provide the detailed algorithm in Appendix A.4. We now derive the resulting higher-level and lower-level gradient formulations.

**Critic LLM Gradient.** We update the higher-level (critic) parameters $\phi$ by ascending the gradient of the Lagrangian

$$\mathcal{L}(\phi,\theta,\lambda) = U(\phi,\theta) + \lambda\big(L(\phi,\theta) - L(\phi,\theta^*(\phi))\big).$$

Let $\tau = (x, y_0, z, y_1)$ denote a trajectory. We denote the distribution induced by the current actor $\theta$ as $\tau \sim \pi_{\theta,\phi}$ and by the optimal actor $\theta^*$ as $\tau^* \sim \pi_{\theta^*,\phi}$. Here, $E_{\tau \sim \pi_{\theta,\phi}}$ denotes the expectation $E_{x\sim\mathcal{D},\, y_0\sim\pi_\theta^L(\cdot|x),\, z\sim\pi_\phi^H(\cdot|x,y_0),\, y_1\sim\pi_\theta^L(\cdot|x,y_0,z)}$. Using the score-function trick and noting that $\theta^*(\phi)$ is treated as fixed inside the partial derivative w.r.t. $\phi$, we obtain:

$$\nabla_\phi \mathcal{L} = (1+\lambda)\, \mathbb{E}_{\tau\sim\pi_{\theta,\phi}}\Big[\nabla_\phi \log \pi_\phi^H(z|x,y_0) R(x,y_1)\Big]$$
$$- \lambda\, \mathbb{E}_{\tau^*\sim\pi_{\theta^*,\phi}}\Big[\nabla_\phi \log \pi_\phi^H(z|x,y_0) R(x,y_1)\Big]. \quad (6)$$

The expectations are estimated via Monte-Carlo rollouts of $(x, y_0, z, y_1)$ and optimized with standard policy-gradient updates; gradients are computed via automatic differentiation. The full derivation is provided in Appendix A.2.

**Actor LLM Gradient.** The gradient of the Lagrangian w.r.t. $\theta$ simplifies as the regularization term vanishes $(\nabla_\theta L(\phi,\theta^*) = 0)$. The lower-level gradient is derived as:

$$\nabla_\theta \mathcal{L} = (1+\lambda)\mathbb{E}_{\tau\sim\pi_{\theta,\phi}}\Big[\Big(\nabla_\theta \log \pi_\theta^L(y_0|x)$$
$$+ \nabla_\theta \log \pi_\theta^L(y_1|x,y_0,z)\Big)R(x,y_1)\Big]. \quad (7)$$

The full derivation is provided in Appendix A.3.

# 4. Experiments

Our empirical analysis addresses the following core research questions to validate our contributions:

- **Q1:** Does Bi-NAC outperform prior fixed and auxiliary feedback-based approaches across model scales?

- **Q2:** Does Bi-NAC outperform GRPO-based reasoning models in terms of performance, parameter and sample efficiency?

- **Q3:** How does reasoning performance scale when extending iterative critic-guided refinement beyond two turns at inference?

- **Q4:** Q4: Does Bi-NAC generate more actionable feedback, and can the actor effectively use it?

- **Q5:** Can Bi-NAC use a single model for both feedback generation and response refinement, and how robust is the framework to heterogeneous pairings?

**Benchmarks and Models.** We evaluate our framework across four diverse reasoning domains:

• MATH-500 , a challenging subset of 500 mathematical problems;
• MBPP, comprising 974 Python programming tasks;
• GPQA, a scientific reasoning benchmark of 3000 graduate-level questions; and
• exceptionally hard competitive math benchmarks, including AIME 2024, AIME 2025, and BEYONDAIME.

The experiments are performed on LLaMA-3.2 (1B, 3B, and 8B) and Qwen-2/Qwen-3 (1.5B, 1.7B, 7B) models. We provide the training efficiency details for Bi-NAC and the baselines in Appendix A.5, rigorous details on experiments and evaluation prompts in Appendix A.6, and the correction instruction prompt templates for the datasets in Appendix A.7.

## 4.1. Main Results

### Q1: Does Bi-NAC outperform prior fixed and auxiliary feedback-based approaches across model scales?

We perform experiments to show that explicitly optimizing the feedback policy drives significantly higher final performance and better convergence speed.

To isolate the contributions of Bi-NAC's bilevel actor-critic alignment, we compare against following baselines: (i) *Behavioral Cloning (BC)* (Torabi et al., 2018), a supervised fine-tuning method using expert trajectories without RL, which enables highlighting the benefits of using RL training with verifiable rewards; (ii) *Hierarchical Non Fine-tuned Zero-Shot (Hier-NFT)*, an approach that we implemented using a pretrained LLM without fine-tuning for both actor and critic, where the higher-level model provides feedback not optimized using verifiable rewards, demonstrating the impact of reward-based fine-tuning; (iii) *ArCHer* (Zhou et al., 2024), an approach that uses an RL trained actor, to assess the advantage of adding RL trained critic; (iv) *SCoRe* (Kumar et al., 2025), an RL based method that fine-

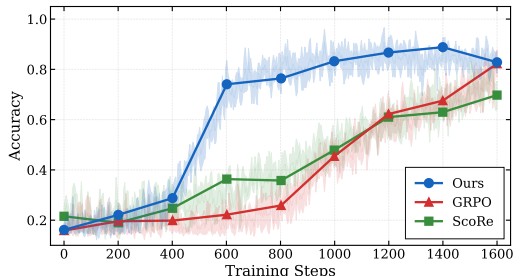

| Method | 1B | | | 3B | | | 8B | | |
|--------|------|------|------|------|------|------|------|------|------|
| | MATH | MBPP | GPQA | MATH | MBPP | GPQA | MATH | MBPP | GPQA |
| BC | 26.2 | 50.0 | 30.2 | 42.4 | 68.5 | 37.1 | 53.0 | 74.5 | 39.0 |
| Hier-NFT | 25.8 | 41.3 | 26.8 | 41.3 | 61.0 | 32.7 | 50.9 | 72.8 | 34.2 |
| ArCHer | 30.6 | 52.4 | 33.6 | 43.6 | 69.5 | 40.9 | 55.5 | 76.0 | 43.0 |
| SCoRe | 39.8 | 62.3 | 34.4 | 45.2 | 70.7 | 42.6 | 56.8 | 77.3 | 44.6 |
| **Bi-NAC** | **46.6** | **66.4** | **40.6** | **51.4** | **75.0** | **49.3** | **60.2** | **79.8** | **56.3** |

*Figure 3.* (**Left**) **Sample Efficiency Comparison.** We compare the training curves for Bi-NAC, GRPO, and SCoRe (Scheurer et al., 2024), and show that Bi-NAC exhibits superior sample-efficiency, faster convergence speed, and higher final accuracy than the baselines. (**Right**) **State-of-the-Art Comparison Across Scales.** We compare Bi-NAC against strong baselines including Behavioral Cloning (BC), Hierarchical-NFT, ArCHer, and SCoRe across three model scales (1B, 3B, and 8B). Bi-NAC consistently outperforms all baselines by significant margins across all benchmarks and scales, demonstrating that our bilevel formulation effectively scales and enhances reasoning capabilities beyond standard reasoning approaches.

tunes the lower-level policy using verifiable rewards but relies on fixed, task-agnostic feedback, to demonstrate the advantages of learning RL fine-tuned feedback over fixed guidance; and (v) auxiliary-critic approaches like *Critique-GRPO* and *Dr. GRPO*, to demonstrate the efficacy of learning RL fine-tuned feedback over auxiliary models.

**Analysis.** As shown in Figure 3 (Left), Bi-NAC translates this aligned feedback into vastly superior sample efficiency and faster convergence speeds compared to GRPO and SCoRe. Figure 3 (Right) confirms that Bi-NAC consistently outperforms all baselines across 1B, 3B, and 8B scales on MATH, MBPP, and GPQA benchmarks. Notably, on the highly challenging GPQA dataset at the 8B scale, Bi-NAC achieves a substantial leap from 44.6% (SCoRe) to 56.3%. As base model capacity increases, Bi-NAC scales more efficiently than standard reasoning approaches, confirming that explicitly training the feedback policy is critical for maximizing reasoning performance across domains. These results show that across domains, Bi-NAC is an effective alternative, particularly in resource-constrained settings where parameter and sample efficiency are critical.

Furthermore, as shown in Figure 4 (Left), adding fixed feedback or auxiliary critics to an RL-trained policy provides only marginal gains. On the MATH-500 benchmark (1B scale), methods relying on an RL-trained actor (ArCHer), fixed feedback (SCoRe), and auxiliary models to generate feedback (Dr. GRPO, Critique-GRPO) yields similarly limited improvements. This performance plateau occurs because fixed and auxiliary critics suffer from actor-critic misalignment. In contrast, Bi-NAC achieves higher accuracy, demonstrating that our bilevel formulation successfully resolves this misalignment and generates high reasoning accuracy.

Finally, to verify that bilevel critic-guided refinement remains effective beyond standard benchmarks, we evaluate Bi-NAC on AIME 2024, AIME 2025, and BeyondAIME

using Qwen3-1.7B (Table 1). While the base model struggles significantly on these tasks, Bi-NAC's aligned feedback mechanism doubles the performance on AIME 2024 (6.7% → 13.3%) and BeyondAIME (2.0% → 4.0%) after just one refinement turn, proving its efficacy on difficult reasoning problems.

| Benchmark | Method | Turn 1 | Turn 2 | Δ |
|-----------|--------|--------|--------|---|
| AIME 2024 | Untrained baseline | 6.7 | 6.7 | 0.0 |
| | **Bi-NAC (Ours)** | 6.7 | **13.3** | **+6.6** |
| AIME 2025 | Untrained baseline | 6.7 | 6.7 | 0.0 |
| | **Bi-NAC (Ours)** | 6.7 | **10.0** | **+3.3** |
| BeyondAIME | Untrained baseline | 2.0 | 2.0 | 0.0 |
| | **Bi-NAC (Ours)** | 2.0 | **4.0** | **+2.0** |

*Table 1.* **AIME benchmark results using Qwen3-1.7B.** Turn 1 shows the base model's initial responses (identical across methods); Turn 2 shows results after one critic-guided refinement step. Bi-NAC's bilevel aligned feedback substantially improves difficult reasoning performance: it doubles accuracy on AIME 2024, AIME 2025, and BeyondAIME, while the untrained baseline shows no improvement.

## Q2. Does Bi-NAC surpass GRPO in performance and parameter efficiency?

In Figure 4 (Right), we show that Bi-NAC outperforms larger GRPO models with superior parameter efficiency. To assess this, we compare smaller Bi-NAC models against larger GRPO models (2B Bi-NAC vs 3B GRPO, and 6B Bi-NAC vs 7B GRPO). In both cases, Bi-NAC outperforms GRPO, highlighting its parameter efficiency. As shown in Figure 4 (Left), 2B Bi-NAC beats 3B GRPO by 5.13 points on MATH, 5.18 on MBPP, and 4.77 on GPQA. The 6B Bi-NAC exceeds the 7B GRPO by 3.0, 2.8, and 5.7 points respectively. These results demonstrate that explicitly aligning the critic not only yields stronger reasoning performance, but also maximizes parameter utilization, making Bi-NAC a highly efficient alternative to scaling up base model sizes.

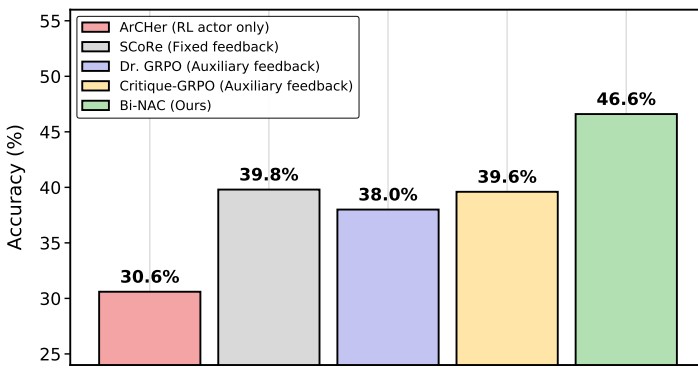

| Method | Size | MATH | MBPP | GPQA |
|--------|------|------|------|------|
| GRPO | 3B | 41.43 | 61.55 | 36.43 |
| **Bi-NAC** | **2B** | **46.56** | **66.73** | **41.20** |
| GRPO | 7B | 48.40 | 72.20 | 43.60 |
| **Bi-NAC** | **6B** | **51.40** | **75.00** | **49.30** |

*Figure 4.* **(Left): Comparison against feedback-based baselines on MATH-500 (1B scale).** Methods that use an RL-trained actor (ArCHer), fixed feedback (SCoRe), or auxiliary critics (Dr. GRPO, Critique-GRPO) give only marginal gains due to actor–critic misalignment; Bi-NAC's bilevel optimization resolves this misalignment and yields substantially higher reasoning accuracy. **(Right): Bi-NAC vs GRPO across model scales.** Bi-NAC is more parameter-efficient—2B Bi-NAC outperforms 3B GRPO, and 6B Bi-NAC outperforms 7B GRPO—showing better accuracy per parameter.

| Method | Architecture | Total Params | Training Time (hrs) | Inference Time (s) | Tokens/ Sample | MATH | MBPP |
|--------|-------------|--------------|---------------------|--------------------|----------------|------|------|
| GRPO | Single actor | 3B | 18.5 | 8.3 | 16,384 | 41.43 | 61.55 |
| Bi-NAC (2B) | 1B actor + 1B critic | 2B | 16.5 | 9.7 | 5,120 | 46.56 | 66.73 |
| Bi-NAC (1B) | Single model | 1B | 16.5 | 9.7 | 5,120 | 46.84 | 65.24 |

*Table 2.* **Compute-performance trade-off.** Comparison of GRPO and Bi-NAC variants showing total parameters, training time (hrs), per-sample inference time (s), tokens per sample, and accuracy on MATH and MBPP. Bi-NAC achieves higher accuracy with fewer total parameters, fewer tokens per sample, and lower training time than GRPO.

Furthermore, as illustrated in Figure 3 (Left), Bi-NAC demonstrates significantly faster convergence compared to standard GRPO. On the MBPP benchmark, GRPO suffers from advantage collapse and improves minimally for the first 600 steps. In contrast, Bi-NAC leverages aligned, task-specific feedback to densify the learning signal. By avoiding this initial zero-reward phase, Bi-NAC accelerates learning and achieves high accuracy in just 400 steps.

We also analyze the compute-performance trade-off to address the perceived computational overhead of training a dual-model architecture. As detailed in Table 2, we compare our 2B Bi-NAC system (1B actor + 1B critic) against a standard 3B GRPO model.

While GRPO relies on exhaustively long reasoning trajectories to find correct answers, consuming 16,384 tokens per sample, Bi-NAC requires only 5,120 tokens per sample. This reduction occurs because Bi-NAC's targeted textual feedback enables the actor to correct errors via shorter, guided rollouts rather than unguided trial and error. Consequently, Bi-NAC achieves 46.56% accuracy on MATH-500 in just 16.5 training hours, whereas the 3B GRPO model requires 18.5 hours to reach only 41.43%. Thus, Bi-NAC demonstrates superior token and sample efficiency, requiring less total training compute to surpass GRPO.

### 4.2. Ablations and Additional Analysis

**Q3: How does reasoning performance scale when extending iterative critic-guided refinement beyond two turns at inference?** Given that Bi-NAC learns an aligned critic during a fixed 2-turn training setup, we investigate whether extending this to iterative multi-turn refinement at inference time yields compounding improvements. We provide a turn-by-turn accuracy analysis on MATH-500 using Qwen3-1.7B in Table 3. The results show that the largest gain occurs from Turn 1 to Turn 2 (+7.8%), with a modest additional improvement at Turn 3 (+1.4%). Crucially, performance remains highly stable for subsequent turns, demonstrating that the model does not degrade with extended inference. This saturation effect justifies our choice of the 2-turn training setup as the optimal compute-efficient configuration, while demonstrating that Bi-NAC's trained critic reliably compounds gains before maintaining performance.

| Method | Turn 1 | Turn 2 | Turn 3 | Turn 4 | Turn 5 |
|--------|--------|--------|--------|--------|--------|
| Untrained Baseline | 62.4 | 65.6 | 62.2 | 64.7 | 63.3 |
| **Bi-NAC (Ours)** | 62.4 | **70.2** | **71.6** | 71.0 | 70.8 |

*Table 3.* **Multi-Turn Accuracy on MATH-500.** Iterative refinement leads to improved performance upto Turn 3.

| Method | Trained Critic | Trained Actor | Accuracy | FO | $\Delta_{\mathrm{acc}}(t_1,t_2)$ | $\Delta_{i \rightarrow c}(t_1,t_2)$ |
|---|---|---|---|---|---|---|
| Hier-NFT | No | No | 35.40 % | 2.1 | +9.60 % | +12.20 % |
| SCoRe | No | Yes | 39.60 % | 2.1 | +7.40 % | +17.80 % |
| Hier-FT | Yes | No | 38.00 % | 3.0 | +8.80 % | +16.00 % |
| Bi-NAC w\o BL | Yes | Yes | 41.80 % | 3.6 | +9.60 % | +19.20 % |
| Bi-NAC | Yes | Yes | **46.60 %** | **4.2** | **+11.40 %** | **+21.20 %** |

*Table 4.* **Feedback actionability and compatibility analysis** We conduct evaluations on MATH-500 with Llama-3.2-1B for various baselines that differ by whether the hierarchical levels are RL-fine-tuned. We report the accuracy, Feedback actionability metric (FO), and two feedback compatibility metrics: $(i)$ accuracy increase from turn 1 to turn 2 $\Delta_{\mathrm{acc}}(t_1,t_2)$, and $(ii)$ the fraction flipped from incorrect to correct $\Delta_{i \rightarrow c}(t_1,t_2)$. Bi-NAC achieves higher accuracy and superior Feedback actionability and feedback compatibility against baselines.

**Q4. In Bi-NAC, does the higher-level policy generate optimal feedback, and can the lower-level policy effectively utilize this feedback?**

Our proposed Bi-NAC framework relies on higher-level feedback to refine lower-level responses, where the performance depends on two key factors: the higher-level policy's ability to generate high-quality, task-relevant feedback, and the lower-level policy's capacity to incorporate this feedback for improved accuracy. To critically evaluate this, we introduce targeted metrics to quantify Feedback actionability and compatibility, and compare Bi-NAC against carefully designed baselines that ablate whether higher-level and lower-level policies are fine-tuned on verifiable rewards.

**Feedback actionability and feedback compatibility metrics.** We define novel metrics to quantitatively assess feedback generation and utilization in Bi-NAC: $(i)$ in the *Feedback actionability* (FO) metric, an LLM judge scores the feedback on a 1–5 scale based on its precision, relevance, and actionability with respect to the input query and first response. Here, the outputs of all methods are passed together to enforce relative grading (see Appendix A.7 for the full evaluation prompt). We also define two *feedback compatibility* metrics, where we compare via: $(ii)$ accuracy increase from initial response to refined response ($\Delta_{acc}(t_1, t_2)$), and $(iii)$ increase in fraction of incorrect $\rightarrow$ correct responses ($\Delta_{i \rightarrow c}(t_1, t_2)$). These metrics evaluate whether a method is able to generate high-quality feedback and leverage it effectively to refine its responses.

**Baselines.** We compare Bi-NAC against multiple baselines that differ by whether higher and lower level policies are RL fine-tuned on verifiable rewards: $(i)$ Hier-NFT (Hierarchical Non Fine-tuned Zero-Shot), where both levels are pre-trained models without RL fine-tuning; $(ii)$ SCoRe, a multi-turn baseline with an RL-fine-tuned lower level but fixed, task-agnostic and generic feedback (e.g., "Please think carefully and generate a correct output."); $(iii)$ Hier-FT (Hierarchical Fine-tuned feedback), where the higher-level policy is RL-trained as in Bi-NAC but the lower level is a pre-trained model; and $(iv)$ Bi-NAC w/o BL, which omits bilevel regularization from the Bi-NAC objective. We use

this baseline to analyze the importance of bi-level optimization in our framework.

**Analysis.** In Table 4, we compare accuracy, Feedback actionability (FO), and feedback compatibility metrics for all baselines. As seen from the table, Bi-NAC consistently achieves the highest accuracies, outperforming baselines and demonstrates superior feedback generation and utilization for refinement. The baselines without higher-level RL fine-tuning, like Hier-NFT and SCoRe, show low FO scores, indicating that pre-trained or fixed feedback lacks the task-specific precision needed for generating aligned feedback, leading to modest compatibility gains. In contrast, Hier-FT with fine-tuned higher-level but pre-trained lower-level policies improves FO but still shows limited compatibility metric values, emphasizing the need for lower-level as well fine-tuning for effective feedback compatibility. Our variant without bilevel regularization (Bi-NAC w/o BL) outperforms earlier baselines but lags behind full Bi-NAC, underscoring the importance of our bi-level optimization in coordinating inter-level dependencies. These results confirm that fine-tuning both levels using our bi-level framework enables Bi-NAC to overcome sparse-reward limitations via dense, actionable feedback, yielding consistent improvements.

**Feedback actionability Analysis.** To analyze whether Bi-NAC is indeed able to generate optimal feedback using hierarchical bi-level RL based formulation, we compare Bilevel-GRPO to Hier-FT (feedback trained while the lower-level policy is kept frozen) and SCoRe (fixed, non-adaptive feedback), using Feedback actionability Score (FO) (Figure 5). Bi-NAC achieves the highest FO score, indicating that it generates near-optimal feedback that more reliably steers the refinement policy toward correct solutions.

**Q5. Can Bi-NAC use a single model for both feedback generation and response refinement, and how robust is the framework to heterogeneous pairings?**

So far, we have discussed Bi-NAC that employs two separate models: one for generating task-specific feedback and another for refining responses conditioned on feedback. Now, we ask: can a single model can handle both tasks?

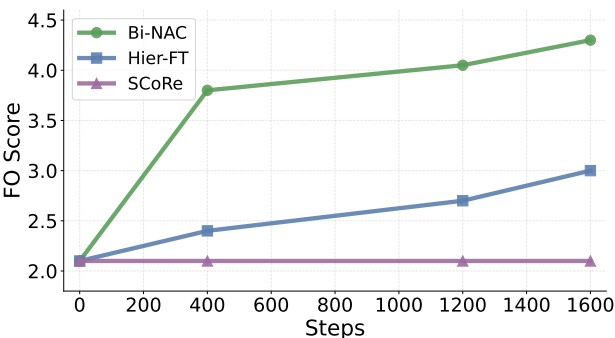

*Figure 5.* Feedback optimality scores across training iterations. Bi-NAC demonstrates superior feedback quality compared to Hier-FT and SCoRe, achieving a 2.1× improvement in FO score after 1600 steps.

This effectively creates a *single-model actor-critic* framework that generates an initial response, generates feedback on it, and then refines its response based on the feedback. Note that this is different from prior thinking tokens or single-level multi-turn based approaches, which do not include a feedback generation step on prior responses for targeted improvements. This single-level variant also minimizes the computational costs compared to two models framework. We implemented and tested a single-model variant of Bi-NAC using the LLaMA-3.2-1B backbone on MATH-500 and MBPP benchmarks. Table 5 shows that the single-model (1B) variant achieves performance comparable to the original two-model (2B) variant, which shows that a single model can effectively generate self-feedback and refine responses, opening future avenues for self-critiquing models. This variant reduces compute costs and simplifies deployment without any performance loss, enhancing Bi-NAC's practicality.

| Method | MATH-500 | MBPP |
|---|---|---|
| Bi-NAC (2M) | 46.56 | 66.73 |
| Bi-NAC (1M) | 46.84 | 65.24 |

Table 5. Comparing Bi-NAC variants: (2M) two models vs. (1M) single model.

**Bi-NAC Heterogeneous Model Pairing Analysis.**

We examine whether Bi-NAC's performance gains stem primarily from using same-family models for the higher and lower-level policies, which could be potentially due to shared architectural features that addresses sparse reward issues. To test this, we evaluate heterogeneous pairings from the LLaMA and Qwen families at the 1B parameter scale. If the gains depended solely on same-family pairings, the performance would degrade in heterogeneous setups. However, as shown in Table 6), the performance remains consistent across all benchmarks for heterogeneous pairings,

confirming that Bi-NAC's main advantages arise from its core framework rather than model family homogeneity.

To demonstrate Bi-NAC's robust performance across cross-family model pairings for higher and lower-level policies, we evaluate using combinations from LLaMA and Qwen families at 1B parameter scale.

| Higher-level | Lower-level | MATH-500 | MBPP | GPQA |
|---|---|---|---|---|
| Qwen-1.5B | Qwen-1.5B | 48.7 | 70.8 | 44.9 |
| LLaMA-1B | Qwen-1.5B | 47.9 | 70.2 | 43.7 |
| Qwen-1.5B | LLaMA-1B | 46.8 | 68.6 | 43.2 |
| LLaMA-1B | LLaMA-1B | 44.6 | 66.7 | 41.2 |

*Table 6.* **Heterogeneous Cross-Family Pairings.** We evaluate Bi-NAC with cross-family pairings of higher and lower-level models across LLaMA and Qwen families on the 1B scale. Our results show that implementing cross-family pairings has minimal effect on the performance, demonstrating that Bi-NAC is able to main performance across various model family combinations.

## 5. Conclusion

We introduced Bi-NAC, a principled bilevel framework that addresses the fundamental misalignment in textual actor-critic methods for reasoning. By formalizing the interaction between critique generation and response refinement as a Stackelberg game, Bi-NAC optimizes feedback for the actor's downstream improvement. Our empirical results across MATH-500, MBPP, and GPQA confirm that this bilevel alignment yields significant gains in sample and parameter efficiency. Notably, Bi-NAC enables smaller models to consistently outperform larger GRPO baselines by effectively leveraging the critic's dense, actionable feedback. We perform an analysis of feedback optimality and compatibility metrics to further validate that our approach produces feedback that is both higher quality and more effectively leveraged by the actor compared to fixed or auxiliary trained baselines. In future work, we plan to explore applying this bilevel framework to multimodal reasoning domains.

## Impact Statement

This work advances the field of reinforcement learning for large language models by introducing a method to train models that generate useful, natural language feedback. By improving how models reason, our approach can make language systems more accurate and interpretable in complex reasoning tasks like math and coding. We encourage continued monitoring to mitigate potential negative applications. These improvements can make language systems more reliable for complex reasoning, but we recommend ongoing monitoring and standard safeguards to reduce misuse and unintended harms.

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

# A. Appendix

## A.1. Related Work

**Reinforcement learning with verifiable rewards for reasoning.** Reinforcement learning with verifiable rewards (RLVR) has recently emerged as a central paradigm for improving the reasoning ability of large language models. In this setting, models are optimized using verifiable outcome signals, such as mathematical correctness, code execution results, or other task-specific verifiers. Group-based policy optimization methods, including GRPO, have been widely adopted because they normalize rewards across multiple sampled responses and avoid training a separate value model (Shao et al., 2024; Guo et al., 2025). Beyond simple outcome-based optimization, recent work shows that bootstrapping reasoning via self-generated rationales enables emergent logical behavior under RLVR (Zelikman et al., 2022). Recent analyses suggest that RLVR can incentivize models to acquire more reliable reasoning patterns during training (Wen et al., 2025; Chen et al., 2025). However, because the supervision is often sparse and terminal, RLVR provides limited credit to partially correct reasoning trajectories. This motivates methods that introduce denser learning signals beyond binary outcome rewards.

**Textual feedback and critique-guided reasoning.** A growing body of work augments scalar rewards with natural language feedback, critiques, or rationales. Early methods include process supervision and step-level feedback, where intermediate reasoning steps are evaluated rather than relying only on final-answer correctness (Lightman et al., 2024; Uesato et al., 2022). Other work uses language models to generate critiques, reflections, or revision instructions that guide self-correction and improve reasoning over multiple attempts (Saunders et al., 2022; Shinn et al., 2023; Madaan et al., 2023; Zhang et al., 2026). More recent RL-based methods incorporate textual feedback directly into policy optimization. For example, Critique-GRPO combines numerical rewards with model-generated critiques and learns from both initial responses and critique-guided refinements (Zhang et al., 2026). Related approaches convert textual critiques into denser optimization signals, allowing natural-language feedback to refine specific parts of model outputs (Wang et al., 2026a), or train models to internalize critique during learning (Song et al., 2026). These methods demonstrate that textual feedback can densify supervision and improve reasoning, but they typically assume that the feedback source is fixed, externally generated, or optimized through objectives not directly tied to the actor's downstream improvement.

**Reward models, generative critics, and limitations of feedback quality.** Another related line of work studies reward models and generative reward models that evaluate or explain the quality of model outputs. Traditional reward models provide scalar preferences or correctness estimates, whereas generative reward models can also produce natural-language rationales or critiques (Wang et al., 2026b). Recent methods include training reward models with synthetic critiques to provide richer, more robust supervisory signals (Ye et al., 2025), and designing generative reward models that explicitly structure their feedback to distinguish between document faithfulness and factual correctness through sequential thinking steps (Ma et al., 2025). This expands the role of the critic from scoring outputs to explaining errors and guiding improvement. However, recent work shows that accurate outcome prediction does not necessarily imply reliable critique generation: a model may assign the correct label while producing flawed or unhelpful explanations (Wang et al., 2026b). Similarly, longer or unconstrained textual feedback can degrade performance (Lloret et al., 2024). These findings suggest that feedback quality cannot be judged solely by fluency, plausibility, or standalone correctness. Therefore, for RL with textual feedback, the key question is whether the feedback is actionable for the actor and improves downstream policy learning.

**Multi-turn self-correction and credit assignment.** Textual feedback is also closely related to multi-turn reasoning and self-correction, where an LLM iteratively critiques, verifies, or revises its own outputs. Methods such as Reflexion, Self-Refine, and ReAct use verbal feedback, internal reflections, or external environment interactions to guide iterative improvement without necessarily optimizing the feedback generator through RL (Shinn et al., 2023; Madaan et al., 2023; Yao et al., 2023). Recent RL formulations model multi-turn reasoning as a sequential decision-making problem and improve credit assignment across turns (Zeng et al., 2025). Other approaches, such as SCoRe, train models to improve their own answers through self-correction using reinforcement learning (Kumar et al., 2025). These works improve the actor's ability to reason over multiple steps, but they do not explicitly optimize a separate textual critic for the actor's future reward after adaptation. In contrast, our work focuses on learning the feedback mechanism itself.

**Bilevel and meta-learning perspectives.** Our work is also related to bilevel optimization and meta-learning, in which one model or objective is optimized via the learning dynamics of another. Bilevel formulations have been used for hyperparameter optimization and intrinsic reward design (Finn et al., 2017; Franceschi et al., 2018). In reinforcement learning, prior work has framed intrinsic reward learning as a bilevel problem, where a learned reward function is optimized to improve downstream policy performance (Stadie et al., 2020). However, these methods typically learn scalar or environment-based reward

functions, rather than natural-language critics that generate textual feedback for LLM reasoning. Our work extends this perspective to RL with textual feedback by formulating the critic-actor interaction as a Stackelberg bilevel program. The critic acts as an adaptive feedback model, optimized for usefulness as measured by the actor's verified reward conditioned on the feedback.

### A.2. Derivation of Higher-level Gradient Objective (Equation 6)

We seek to derive the gradient of the Lagrangian $\mathcal{L}(\phi, \theta, \lambda)$ with respect to the higher-level policy parameters $\phi$. Recall the Lagrangian formulation:

$$\mathcal{L}(\phi, \theta, \lambda) = U(\phi, \theta) + \lambda\big(L(\phi, \theta) - L(\phi, \theta^*(\phi))\big). \tag{8}$$

We need to derive the gradient of this Lagrangian formulation with respect to higher level parameters $\phi$:

$$\nabla_\phi \mathcal{L}(\phi, \theta, \lambda) = \nabla_\phi U(\phi, \theta) + \lambda \nabla_\phi L(\phi, \theta) - \lambda \nabla_\phi L(\phi, \theta^*(\phi)). \tag{9}$$

We start with the higher-level objective function defined as:

$$U(\phi, \theta) = \mathbb{E}_{x \sim \mathcal{D}, \, y_0 \sim \pi_\theta^L(\cdot|x), z \sim \pi_\phi^H(\cdot|x,y_0), y_1 \sim \pi_\theta^L(\cdot|x,y_0,z)} \left[ R(x, y_1) \right]. \tag{10}$$

We first derive the gradient for the objective $\nabla_\phi U(\phi, \theta)$ by expanding its expectation formulation into a summation:

$$\nabla_\phi U(\phi, \theta) = \nabla_\phi \sum_{x, y_0, z, y_1} p(x) \pi_\theta^L(y_0|x) \pi_\phi^H(z|x, y_0) \pi_\theta^L(y_1|x, y_0, z) R(x, y_1) \tag{11}$$

$$= \sum_{x, y_0, z, y_1} p(x) \pi_\theta^L(y_0|x) \left[ \nabla_\phi \pi_\phi^H(z|x, y_0) \right] \pi_\theta^L(y_1|x, y_0, z) R(x, y_1). \tag{12}$$

Using the log-derivative trick $\nabla_\phi \pi_\phi^H = \pi_\phi^H \nabla_\phi \log \pi_\phi^H$, we obtain:

$$\nabla_\phi U(\phi, \theta) = \mathbb{E}_{x \sim \mathcal{D}, \, y_0 \sim \pi_\theta^L(\cdot|x), z \sim \pi_\phi^H(\cdot|x,y_0), y_1 \sim \pi_\theta^L(\cdot|x,y_0,z)} \left[ \nabla_\phi \log \pi_\phi^H(z \mid x, y_0) \cdot R(x, y_1) \right]. \tag{13}$$

Since $U(\phi, \theta)$ and $L(\phi, \theta)$ share the same expectation structure (expected reward over trajectories generated by $\pi_\phi^H$ and $\pi_\theta^L$), we can use the above steps to derive $L(\phi, \theta)$ to get the formulation:

$$\nabla_\phi L(\phi, \theta) = \mathbb{E}_{x \sim \mathcal{D}, \, y_0 \sim \pi_\theta^L(\cdot|x), z \sim \pi_\phi^H(\cdot|x,y_0), y_1 \sim \pi_\theta^L(\cdot|x,y_0,z)} \left[ \nabla_\phi \log \pi_\phi^H(z \mid x, y_0) \cdot R(x, y_1) \right]. \tag{14}$$

Substituting $\nabla_\phi U(\phi, \theta)$ and $\nabla_\phi L(\phi, \theta)$ back into Equation 9, we get the final update rule for $\phi$:

$$\nabla_\phi \mathcal{L}(\phi, \theta, \lambda) = (1 + \lambda) \mathbb{E}_{x \sim \mathcal{D}, \, y_0 \sim \pi_\theta^L(\cdot|x), z \sim \pi_\phi^H(\cdot|x,y_0), y_1 \sim \pi_\theta^L(\cdot|x,y_0,z)} \left[ \nabla_\phi \log \pi_\phi^H(z \mid x, y_0) \cdot R(x, y_1) \right] \tag{15}$$

$$- \lambda \mathbb{E}_{x \sim \mathcal{D}, \, y_0 \sim \pi_{\theta^*(\phi)}^L(\cdot|x), z \sim \pi_\phi^H(\cdot|x,y_0), y_1 \sim \pi_{\theta^*(\phi)}^L(\cdot|x,y_0,z)} \left[ \nabla_\phi \log \pi_\phi^H(z \mid x, y_0) \cdot R(x, y_1) \right]. \tag{16}$$

This update encourages the critic to generate feedback that maximizes reward for the *current* actor ($\theta$), while being regularized by the performance of the *optimal* (or reference) actor ($\theta^*(\phi)$). □

### A.3. Derivation of Lower-level Gradient Objective (Equation 7)

We seek to derive the gradient of the Lagrangian $\mathcal{L}(\phi, \theta, \lambda)$ with respect to the lower-level policy parameters $\theta$. Recall the Lagrangian formulation:

$$\mathcal{L}(\phi, \theta, \lambda) = U(\phi, \theta) + \lambda\big(L(\phi, \theta) - L(\phi, \theta^*(\phi))\big). \tag{17}$$

Taking the gradient with respect to $\theta$:

$$\nabla_\theta \mathcal{L}(\phi, \theta, \lambda) = \nabla_\theta U(\phi, \theta) + \lambda \nabla_\theta L(\phi, \theta) - \lambda \underbrace{\nabla_\theta L(\phi, \theta^*(\phi))}_{=0}. \tag{18}$$

The third term vanishes because $L(\phi, \theta^*(\phi))$ depends on $\theta^*$ (which is a function of $\phi$), not on the current parameters $\theta$. Thus, the gradient simplifies to:

$$\nabla_\theta \mathcal{L}(\phi, \theta, \lambda) = (1 + \lambda)\nabla_\theta L(\phi, \theta). \tag{19}$$

(Note: Since $U(\phi, \theta)$ and $L(\phi, \theta)$ are identical functions, both representing the expected reward $\mathbb{E}[R]$. Hence, their gradients with respect to $\theta$ have similar formulation).

We now derive $\nabla_\theta L(\phi, \theta)$. Expanding the expectation into a summation:

$$L(\phi, \theta) = \sum_{x, y_0, z, y_1} p(x)\pi_\theta^L(y_0|x)\pi_\phi^H(z|x, y_0)\pi_\theta^L(y_1|x, y_0, z)R(x, y_1). \tag{20}$$

Applying the gradient $\nabla_\theta$, we apply the product rule to the two terms dependent on $\theta$: $\pi_\theta^L(y_0|x)$ and $\pi_\theta^L(y_1|x, y_0, z)$:

$$\nabla_\theta L(\phi, \theta) = \sum_{x, y_0, z, y_1} p(x)\pi_\phi^H(z|x, y_0)R(x, y_1)\Big[\nabla_\theta \pi_\theta^L(y_0|x)\pi_\theta^L(y_1|x, y_0, z) + \pi_\theta^L(y_0|x)\nabla_\theta \pi_\theta^L(y_1|x, y_0, z)\Big]. \tag{21}$$

Using the log-derivative trick $\nabla_\theta \pi = \pi \nabla_\theta \log \pi$:

$$\nabla_\theta L(\phi, \theta) = \sum_{x, y_0, z, y_1} p(x)\pi_\theta^L(y_0|x)\pi_\phi^H(z|x, y_0)\pi_\theta^L(y_1|x, y_0, z)R(x, y_1)\cdot\Big[\nabla_\theta \log \pi_\theta^L(y_0|x) + \nabla_\theta \log \pi_\theta^L(y_1|x, y_0, z)\Big]. \tag{22}$$

Rewriting as an expectation:

$$\nabla_\theta L(\phi, \theta) = \mathbb{E}_{x\sim\mathcal{D},\, y_0\sim\pi_\theta^L(\cdot|x), z\sim\pi_\phi^H(\cdot|x,y_0), y_1\sim\pi_\theta^L(\cdot|x,y_0,z)}\Big[\big(\nabla_\theta \log \pi_\theta^L(y_0|x) + \nabla_\theta \log \pi_\theta^L(y_1|x, y_0, z)\big)\cdot R(x, y_1)\Big]. \tag{23}$$

Substituting back into the Lagrangian gradient:

$$\nabla_\theta \mathcal{L}(\phi, \theta, \lambda) = (1+\lambda)\mathbb{E}_{x\sim\mathcal{D},\, y_0\sim\pi_\theta^L(\cdot|x), z\sim\pi_\phi^H(\cdot|x,y_0), y_1\sim\pi_\theta^L(\cdot|x,y_0,z)}\Big[\big(\nabla_\theta \log \pi_\theta^L(y_0|x) \tag{24}$$

$$+ \nabla_\theta \log \pi_\theta^L(y_1|x, y_0, z)\big)\cdot R(x, y_1)\Big]. \tag{25}$$

This result shows that the lower-level update is simply a standard policy gradient update on the actor, scaled by the Lagrange multiplier term $(1 + \lambda)$. $\qquad\square$

### A.4. Bi-NAC Algorithm

Here, we provide the complete algorithm for Bi-NAC

---

**Algorithm 1** Bi-NAC: Bilevel Natural Language Actor-Critic

---

1: **Initialize:** Critic parameters $\phi$, Actor parameters $\theta$.
2: **Hyperparameters:** Lagrangian multiplier $\lambda$, learning rates $\alpha_\theta, \alpha_\phi$.
3: **for** each iteration **do**
4:    **# 1. Update Lower-Level Actor Policy ($\theta$)** {Maximizes task reward given current feedback (Eq. 7)}
5:    **for** each actor update step **do**
6:       Sample batch of prompts $x \sim \mathcal{D}$
7:       Generate trajectory:
8:          $y_0 \sim \pi_\theta^L(\cdot|x)$ {Initial attempt}
9:          $z \sim \pi_\phi^H(\cdot|x, y_0)$ {Critic feedback}
10:         $y_1 \sim \pi_\theta^L(\cdot|x, y_0, z)$ {Refined response}
11:       Compute reward $R(x, y_1)$
12:       Estimate Gradient $\nabla_\theta \mathcal{L}$ (Eq. 7):
13:         $g_\theta \leftarrow (1 + \lambda) \cdot R(x, y_1) \cdot \nabla_\theta \left( \log \pi_\theta^L(y_0|x) + \log \pi_\theta^L(y_1|x, y_0, z) \right)$
14:       Update $\theta \leftarrow \theta + \alpha_\theta \cdot g_\theta$
15:    **end for**
16:    **# 2. Update Higher-Level Critic Policy ($\phi$)** {Maximizes actor's improvement (Eq. 6)}
17:    **for** each critic update step **do**
18:       Sample batch of prompts $x \sim \mathcal{D}$
19:       *// Term 1: Current Actor Rollout ($\tau \sim \pi_{\theta,\phi}$)*
20:       Generate $\tau = (x, y_0, z, y_1)$ using current $\theta, \phi$
21:       $G_1 \leftarrow (1 + \lambda) \cdot R(x, y_1) \cdot \nabla_\phi \log \pi_\phi^H(z|x, y_0)$
22:       *// Term 2: "Optimal" Actor Rollout ($\tau^* \sim \pi_{\theta^*,\phi}$)*
23:       {Approximate $\theta^*$ via recent checkpoint k to get $\theta^k$}
24:       Generate $\tau^* = (x, y_0^*, z, y_1^*)$ using $\theta^*, \phi$, where we get $y_0^*$ and $y_1^*$ from $\theta^k$
25:       $G_2 \leftarrow \lambda \cdot R(x, y_1^*) \cdot \nabla_\phi \log \pi_\phi^H(z|x, y_0^*)$
26:       Update $\phi \leftarrow \phi + \alpha_\phi \cdot (G_1 - G_2)$
27:    **end for**
28: **end for**

---

## A.5. Training Efficiency Analysis

We conducted a comprehensive training efficiency analysis on the mathematical reasoning task, comparing our proposed method Bi-NAC against baseline methods including Behavioral Cloning (BC), SCoRe, ArCHer and GRPO. Experiments were performed using a 1B parameter model with a rollout size of 128 for 70 training iterations. The training was executed on an NVIDIA L40S GPU equipped with 48GB memory, with an average power usage of approximately 280W (training and idle weighted average). Table 7 summarizes key efficiency metrics: training time (in hours), GPU memory usage (in GB), estimated $CO_2$ emissions per training run (in kilograms), achieved TFLOPs per second, and the average inference time in seconds (averaged over 100 prompts).

*Table 7.* Training efficiency and inference performance comparison of different methods on the mathematical reasoning task (1B model, rollout size 128, 70 iterations).

| Method | Training Time (hrs) | GPU Memory (GB) | $CO_2$ Emissions (kg) | TFLOPs/sec | Inference Time (s) |
|--------|---------------------|-----------------|-----------------------|------------|---------------------|
| BC | 1.2 | 8 | 0.14 | 132 | 4.5 |
| ArCHer | 22.1 | 28 | 2.46 | 91 | 7.9 |
| SCoRe | 13.6 | 15 | 1.52 | 107 | 8.0 |
| GRPO | 18.5 | 24 | 2.05 | 95 | 8.3 |
| Bi-NAC | 16.5 | 22 | 1.83 | 83 | 9.7 |

These results highlight the significantly higher computational and memory demands of RL-based and hierarchical approaches compared to supervised behavioral cloning (BC). Specifically, Bi-NAC and ArCHer require the longest training times and the largest GPU memory footprints. Despite this increased cost, Bi-NAC achieves a favorable accuracy-compute trade-off

by delivering substantial performance improvements over both simple supervised and existing hierarchical baselines.

### A.6. Experimental Details

Experiments are conducted by fine-tuning various models: LLAMA-3.2-1B-INSTRUCT, LLAMA-3.2-3B-INSTRUCT LLAMA-3.1-8B-INSTRUCT, QWEN2-1.5B-INSTRUCT and QWEN2-7B-INSTRUCT. These models are trained using Low-Rank Adaptation (LoRA) with rank $r = 16$, a LoRA alpha of 32, and a dropout of 0.05. The value critic in hierarchical approaches employs a DISTILROBERTA-BASE encoder architecture. Models are trained with a rollout size of 128 for 70 total iterations, split into 35 iterations for Turn 1 and 35 iterations for Turn 2 wherever applicable for multi-turn methods. For the LLAMA-3.2-3B model, we used a smaller rollout size of 32 and trained for 10 total iterations. For methods employing multi-turn training (e.g., SCORE, BI-NAC), the first and second-turn trajectories are logged to facilitate off-policy updates in Turn 2. Across all experiments, we use a decoding temperature of 0.7. Detailed hyperparameters, including specific learning rates and batch sizes for each domain, are provided in Appendix Table 8. We provide more examples comparing BI-NAC's reasoning on questions from the MATH, HUMANEVAL and GPQA benchmarks against GRPO and SCORE in Appendix A.8.

For mathematical reasoning, final outputs are extracted and judged via DEEPEVAL evaluator that uses an O3-MINI judge for algebraic equivalence. For code, programs are executed in a secure sandbox against test cases. We provide training efficiency comparisons in Appendix Table 7, showing that Bi-NAC uses fewer resources than baselines like GRPO and ArCHer (Zhou et al., 2024), while achieving superior performance. See Appendix Sec A.5 for detailed analysis.

In Table 8, we provide the detailed hyperparameters, like specific learning rates and batch sizes for each domain on experiments conducted by fine-tuning LLAMA-3.2 1B-INSTRUCT and LLAMA-3.2 3B-INSTRUCT models. These models are trained using Low-Rank Adaptation (LoRA) with rank $r = 16$, LoRA alpha $\alpha = 32$, and a dropout of $p = 0.05$. The value critic in hierarchical approaches employs a pretrained DISTILROBERTA-BASE encoder architecture. Models are trained with a rollout size of 128 for 70 total iterations, split into 35 iterations each for Stage I and Stage II in two-stage methods. For the LLAMA-3.2 3B model, we use a smaller rollout size of 32 and train for only 10 total iterations due to memory constraints. Across all experiments, we use a decoding temperature of $\tau = 0.7$. The configuration choices were validated through ablation studies on held-out validation splits.

Table 8. Hyperparameters used for Bi-NAC experiments with LLAMA-1B and LLAMA-3B backbones.

| Setting | 1B | 3B |
| --- | --- | --- |
| LoRA Rank | 16 | 16 |
| LoRA Alpha | 32 | 32 |
| LoRA Dropout | 0.05 | 0.05 |
| Actor Learning Rate | 5e-6 | 5e-6 |
| Critic Learning Rate | 1e-4 | 1e-4 |
| Optimizer | Adam | Adam |
| Batch Size | 128 | 64 |
| Iterations (Total) | 1600 | 1600 |
| Iterations (Stage I / II) | 600/1000 | 600/1000 |
| Value Critic Model | DistilRoBERTa | DistilRoBERTa |
| Decoding Temperature | 0.7 | 0.7 |

**Evaluation prompts.** We use prompt templates as employed in the SCoRe paper (Kumar et al., 2025) for the flat policies, i.e., the single-turn approaches. These templates are adapted to each dataset: (1) a zero-shot chain-of-thought format for MATH-500 (Wei et al., 2022), and (2) a canonical $K$-shot format with $K = 3$ for HumanEval (Chen et al., 2021). We designed these templates to elicit high-quality, task-relevant responses from the model, while following prompting strategies established in prior state-of-the-art work(Brown et al., 2020). For two-turn methods, although we extend the same base templates as above for the first turn, for the second turn, we prepend a brief but explicit task guidance instruction for the higher-level policy to direct the model to generate the guidance instruction. This guidance is then passed to the lower-level policy to generate refined outputs.

## A.7. Prompts

Effective prompts are important for eliciting high-quality, task-aligned responses in LLMs. Building on the template framework, we design dataset-specific prompts for both initial generations and iterative refinements:

- **MATH-500**: Zero-shot chain-of-thought prompts encourage stepwise reasoning while preserving generalization to unseen problems.

- **HumanEval**: Canonical 3-shot templates provide in-context examples of input-output pairs, aligning with established code generation benchmarks.

For hierarchical self-correction, we extend these templates by prepending explicit guidance in the second turn. This structured intervention—unique to Bi-NAC—directs the model's attention to specific error patterns while maintaining coherence with the original task context.

Full prompt templates, including first-turn instructions and second-turn correction guidance, are provided below. Our design balances reproducibility (via reuse of SCoRe's templates) with innovation (via guided error localization), enabling systematic self-correction across diverse domains.

**MATH-500 Prompts**

MATH-500: Zero-shot Prompt

```
You are a math expert. When you respond, respond only with the Solution of the final
    Problem, thinking step by step. At the end of the Solution, when you give your final
    answer, write it in the form "Final Answer: The final answer is $answer$. I hope it is
     correct."
```

MATH-500: Task-Agnostic Guidance Instruction

```
There might be an error in the solution above because of lack of understanding of the
    question. Please correct the error, if any, and rewrite the solution. Only output the
    final solution! At the end of the Solution, when you give your final answer, write it
    in the form "Final Answer: The final answer is $answer$. I hope it is correct."
```

MATH-500: Guidance Generation Prompt

```
You are an expert math tutor reviewing a student's solution to a math problem.

PROBLEM:
{problem}

INITIAL SOLUTION:
{solution}

PROMPT: First, analyze the solution for errors or misconceptions. Then, write a brief,
    helpful instruction that will guide the student toward correcting their solution. Your
     instruction should be specific to the errors you identified, but don't solve the
    problem for them. Your response should be ONLY the instruction for the student to
    improve their solution, nothing else. DO NOT include ANY SOLUTION.

GUIDING INSTRUCTION:
```

MATH-500: Task Specific Guidance Prompt

```
{problem}
{solution}

Suggestive Correction:

{custom_instruction}
```

## HumanEval Prompts

### HumanEval: Zero-shot Prompt

```
You are an expert programmer. Below is a programming problem. Write a solution in {
    language}.
Make sure your solution is correct, efficient, and addresses all the requirements of the
    problem.
When you're done, wrap your code in triple backticks with the language specified, like:
    ```{language} (your code here) ```

Problem:
{prompt}

Solution:
```

### HumanEval: Task-Agnostic Guidance Instruction

```
Your code might have issues or bugs, or it may not be optimized. Please review your
    solution, identify any problems, and provide an improved solution.
Make sure your solution passes all test cases and meets all requirements. Remember to wrap
     your code in triple backticks with the language specified, like: ```{language} (your
    code here) ```
```

### HumanEval: Guidance Generation Prompt

```
You are an expert programming mentor reviewing code written by a student.
PROBLEM:
{problem}
STUDENT'S SOLUTION:
{solution}
PROMPT: First, analyze the solution for bugs, inefficiencies, or edge cases it doesn't
    handle. Then, write a brief, helpful instruction that will guide the student toward
    correcting their solution.
Your instruction should be specific to the issues you identified, but don't solve the
    problem completely for them.
Your response should be ONLY the instruction for the student to improve their solution,
    nothing else. DO NOT write any code.

GUIDING INSTRUCTION:
```

### HumanEval: Task Specific Guidance Prompt

```
{problem}
{solution}

Code Review Feedback:

{custom_instruction}

Please fix these issues and provide an improved solution. Remember to wrap your code in
    triple backticks with the language specified, like: ```{language} (your code here) ```
```

## Feedback Optimality (FO) GEval Prompt

### Feedback Optimality Multi-Method Prompt

```
You are an expert evaluator. Your task is to assess how optimal each method's
feedback is with respect to the same input query and incorrect answer.

"Optimal" means the feedback:
- is precise and directly relevant to the query,
- identifies key mistakes in the incorrect answer,
- provides actionable guidance to refine towards the correct answer,
```

```
- avoids vague, generic, or misleading suggestions.

Scoring Rubric (1 to 5):
1 = Irrelevant or misleading; no useful guidance.
2 = Weak or vague; somewhat related but mostly unhelpful.
3 = Fair; partially useful but incomplete or missing critical aspects.
4 = Good; mostly relevant, helpful, with minor gaps.
5 = Optimal; precise, actionable, and directly enables correction.

You will receive:
- Input Query
- Incorrect Answer
- Feedback from multiple methods

For each method, return only the method name and a score (1 to 5).
Do not explain your reasoning. Do not add extra text.

---

Input Query:
{query}

Incorrect Answer:
{incorrect_answer}

Feedback from Methods:
ZSCoT: {feedback_zscot}

SCoRe-FF: {feedback_score_ff}

TF-LBM: {feedback_tf_lbm}

GRPO: {feedback_grpo}

Bi-NAC w/o bilevel: {feedback_bi_nac_wo_bilevel}

Bi-NAC: {feedback_bi_nac}

---

Output format (strictly JSON):
{
  "ZSCoT": <score>,
  "SCoRe-FF": <score>,
  "TF-LBM": <score>,
  "GRPO": <score>,
  "Bi-NAC w/o BL": <score>,
  "Bi-NAC": <score>
}
```

### A.8. Illustrative Examples of BI-NAC

Here, we provide a few examples on the MATH, HumanEval and GPQA datasets. Whereas prior single-level multi-turn RL approaches like Score fail to generate correct outputs, Bi-NAC leverages aligned actor-critic models solve complex tasks.

**MATH Example 1**

**Prompt:** If $2^8 = 4^x$, find the value of $x$.

**Bi-NAC**

**1. First response ($y_1$) (Incorrect):**
Here also the solver logs both sides, simplifies to $8 \log 2 = x \log 4$, uses $\log 4 = 2 \log 2$, and divides to get $x = 8$.

**Task-Specific Guidance ($z_l$)**

1. Rewrite $4$ as $2^2$ so that both sides share the same base.
2. Apply the power-of-a-power rule to obtain $2^8 = 2^{2x}$.
3. Equate exponents ($8 = 2x$) and solve for $x$.

**2. Improved response ($y_2$) (Correct):**
By rewriting $2^8 = (2^2)^x$ to get $2^8 = 2^{2x}$, equating exponents gives $8 = 2x$, hence $x = 4$.

**MATH Example 2**

**Prompt:** Let
$$f(n) = \left\{ \begin{array}{ll} \lfloor n \rfloor & \text{if } n \geq 4, \\ \lceil n \rceil & \text{if } n < 4. \end{array} \right\}.$$
Find $f\left(\frac{\pi}{3}\right) + f(\sqrt{45}) + f(8^{2/3})$.

**Bi-NAC**

// Step 1: Evaluate the function f(n) for each given input
For $\frac{\pi}{3}$, since $\frac{\pi}{3} \geq 4$, $f\left(\frac{\pi}{3}\right) = \lfloor \frac{\pi}{3} \rfloor = 1$. For $\sqrt{45}$, since $\sqrt{45} < 4$, $f(\sqrt{45}) = \lceil \sqrt{45} \rceil = 6$. For $8^{2/3}$, since $8^{2/3} \geq 4$, $f(8^{2/3}) = \lfloor 8^{2/3} \rfloor = 2$.
// Step 2: Add the values of f(n) for each input $f\left(\frac{\pi}{3}\right) + f(\sqrt{45}) + f(8^{2/3}) = 1 + 6 + 2 = 9$
The final answer is: $\boxed{9}$.

**Task-Specific Guidance ($z_l$)**

// Step 1: Evaluate the function f(n) for each given input Understand that the function definition depends on whether the input is greater than or less than 4. Use ceiling if the input is $< 4$, and floor if it's $\geq 4$.
// Step 2: Classify the inputs Check whether each of the values — $\pi/3$, $\sqrt{(45)}$, and $8^{2/3}$ — is greater than or less than 4.
// Step 3: Apply the appropriate function For inputs $< 4$, use the ceiling function; for inputs $\geq 4$, use the floor function. Be sure to compute accurate decimal approximations if needed.
// Step 4: Add the results Once each value of f(n) is computed correctly, sum them up to find the final result.

**2. Improved response ($y_2$) (Correct):**
// Step 1: Evaluate $f\left(\frac{\pi}{3}\right)$. Since $\frac{\pi}{3} \approx 1.047$, we have $f\left(\frac{\pi}{3}\right) = \lceil \frac{\pi}{3} \rceil = 1$.
// Step 2: Evaluate $f(\sqrt{45})$. Since $\sqrt{45} \approx 6.708$, we have $f(\sqrt{45}) = \lceil \sqrt{45} \rceil = 7$.
// Step 3: Evaluate $f(8^{2/3})$. Since $8^{2/3} = 4$, we have $f(8^{2/3}) = \lfloor 8^{2/3} \rfloor = 4$.
// Step 4: Calculate the final result We have $f\left(\frac{\pi}{3}\right) + f(\sqrt{45}) + f(8^{2/3}) = 1 + 7 + 4 = 12$.
The final answer is: $\boxed{12}$.

---

### MATH Example 3

**Prompt:** Let $z = 1 + i$ and $w = \dfrac{3z+1}{5z+7}$. Find $|w|$.

---

#### Bi-NAC

**1. First response** ($y_1$) **(Incorrect):**

// Step 1: Calculate the value of $z + w$. We need to calculate the value of $z + w$. We know that $z = 1 + i$ and $w = \dfrac{3z+1}{5z+7}$. Therefore, we can substitute values of $z$ and $w$ into the equation and simplify.

// Step 2: Substitute values of $z$ and $w$ into $z + w = (1 + i) + \dfrac{3(1+i)+1}{5(1+i)+7} = (1 + i) + \dfrac{4+3i}{12+5i}$

// Step 3: Simplify the expression We can multiply the numerator and denominator by the conjugate of the denominator, which is $12 - 5i$. This will eliminate the imaginary part from the denominator.

// Step 4: Multiply the numerator and denominator by the conjugate of the denominator $(1 + i) + \dfrac{(4+3i)(12-5i)}{(12+5i)(12-5i)} = (1 + i) + \dfrac{48 - 20i + 36i - 15i^2}{144 + 25}$

// Step 5: Simplify the numerator and denominator Since $i^2 = -1$, we can simplify the numerator and denominator. $(1 + i) + \dfrac{48 - 20i + 36i + 15}{169} = (1 + i) + \dfrac{63 + 16i}{169}$

// Step 6: Simplify the fraction $(1 + i) + \dfrac{63 + 16i}{169} = \dfrac{169 + 169i + 63 + 16i}{169} = \dfrac{232 + 185i}{169}$

// Step 7: Calculate the absolute value $|w| = \left| \dfrac{232 + 185i}{169} \right| = \sqrt{\left(\dfrac{232}{169}\right)^2 + \left(\dfrac{185}{169}\right)^2}$. Answer: $\boxed{\dfrac{291}{169}}$.

---

#### Task-Specific Guidance ($z_i$)

// Step 1: Calculate the value of z + w Recall how to define z and use it to express w in terms of z. You'll need to simplify a complex fraction.

// Step 2: Substitute the values Carefully substitute z = 1 + i into the expression for w. Pay attention to both the real and imaginary parts when expanding.

// Step 3: Simplify the expression Group the real and imaginary terms in the numerator and denominator to prepare for rationalizing the denominator.

// Step 4: Multiply by the conjugate To simplify a complex fraction, multiply the top and bottom by the conjugate of the denominator. Remember what the conjugate of a complex number looks like.

// Step 5: Find the magnitude of w To find the absolute value (magnitude) of a complex number, apply the formula sqrt(a² + b²) using the real and imaginary parts of w.

---

**2. Improved response** ($y_2$) **(Correct):**

// Step 1: Define the variables Let $z = 1 + i$ and $w = \dfrac{3z+1}{5z+7}$.

// Step 2: Simplify the expression for w To simplify the expression for w, we need to substitute the value of z into the expression for w.

// Step 3: Substitute the value of z into the expression for w $w = \dfrac{3(1+i)+1}{5(1+i)+7}$

// Step 4: Simplify the numerator of the expression for w $3(1+i) + 1 = 3 + 3i + 1 = 4 + 3i$

// Step 5: Simplify the denominator of the expression for w $5(1+i) + 7 = 5 + 5i + 7 = 12 + 5i$

// Step 6: Substitute the numerator and denominator back into expression for w $w = \dfrac{4+3i}{12+5i}$

// Step 7: Multiply the numerator and denominator of the expression for w by the conjugate of the denominator To simplify the expression for w, we need to multiply the numerator and denominator by the conjugate of the denominator.

// Step 8: Multiply numerator and denominator by conjugate of denominator $w = \dfrac{(4+3i)(12-5i)}{(12+5i)(12-5i)}$

// Step 9: Simplify the numerator and denominator $w = \dfrac{48 - 20i + 36i - 15i^2}{144 - 25i^2}$

// Step 10: Simplify the numerator and denominator using the fact that $i^2 = -1$ $w = \dfrac{48 + 16i + 15}{144 + 25}$

// Step 11: Simplify the square root expression $|w| = \sqrt{\dfrac{4225}{28561}} = \dfrac{65}{169}$. The final answer is: $\boxed{\dfrac{65}{169}}$.

---

### HumanEval Example 1

**Prompt:**
from typing import List
def filter_by_prefix(strings: List[str], prefix: str) -¿ List[str]:
""" Filter an input list of strings only for ones that start with a given prefix.
¿¿¿ filter_by_prefix([], 'a')
[]
¿¿¿ filter_by_prefix(['abc', 'bcd', 'cde', 'array'], 'a')
['abc', 'array']
""".

---

### Bi-NAC

**1. First response ($y\_1$) (Incorrect):**
from typing import List
def filter_by_prefix(strings: List[str], prefix: str) -¿ List[str]:
"""

Filter an input list of strings only for ones that start with a given prefix.
Args:
strings (List[str]): A list of strings to be filtered.
prefix (str): The prefix to filter by.
Returns:
List[str]: A list of strings that start with the given prefix.
"""

print(filter_by_prefix(strings, prefix))
# Output: ['abc', 'array']

#### Task-Specific Guidance ($z_l$)

The student's filter_by_prefix function does not handle the case where the input list is empty. They should add a check at the beginning of the function to return an empty list in this case. Additionally, they should consider the case where the prefix is an empty string. They should add a check at the beginning of the function to return an empty list in this case as well. Finally, they should consider the case where the input list contains strings that are not strings. They should add a check at the beginning of the function to raise a TypeError if the input is not a list or if the prefix is not a string.

**2. Improved response ($y\_2$) (Correct):**
from typing import List def filter_by_prefix(strings: List[str], prefix: str) -¿ List[str]:
"""

Filter an input list of strings only for ones that start with a given prefix.
Args:
strings (List[str]): A list of strings to be filtered.
prefix (str): The prefix to filter by.
Returns:
List[str]: A list of strings that start with the given prefix.
"""

# Check if the input is a list
if not isinstance(strings, list):
raise TypeError("Input must be a list")
# Check if the prefix is a string
if not isinstance(prefix, str):
raise TypeError("Prefix must be a string")
# Check if the input list is empty
if not strings:
return []
# Use list comprehension to create a new list containing only the strings that start with the prefix return [s for s in strings if s.startswith(prefix)].

---

**GPQA Example 1**

---

**Prompt:** Two quantum states with energies $E_1$ and $E_2$ have a lifetime of $10^{-9}$ sec and $10^{-8}$ sec, respectively. We want to clearly distinguish these two energy levels. Which one of the following options could be their energy difference so that they can be clearly resolved?

**Bi-NAC**

**1. First response ($y_1$) (Incorrect):**
"Take smaller lifetime $\tau = 10^{-9}$ s, $\Delta E \approx \hbar/\tau \sim 10^{-9}$ eV."

> **Task-Specific Guidance ($z_l$)**
>
> Use the energy–time uncertainty relation $\Delta E \Delta t \sim \hbar$ for each state. To distinguish, the energy separation must exceed the larger of the uncertainty widths. Compute $\Delta E$ using $\tau = 10^{-8}$ and $\tau = 10^{-9}$, then identify a feasible resolvable difference.

**2. Improved response ($y_2$) (Correct):**
$\Delta E \sim \hbar/\tau$. For $\tau = 10^{-9}$ s, $\Delta E \approx 6.6 \times 10^{-7}$ eV; for $\tau = 10^{-8}$ s, $\Delta E \approx 6.6 \times 10^{-8}$ eV. To clearly resolve, energy difference must be larger than both $\rightarrow$ of order $10^{-4}$ eV (realistic discrete choice).

The final answer is: $\boxed{10^{-4} \text{ eV}}$.

---

**GPQA Example 2**

---

**Prompt:** A light beam is propagating through a glass with index of refraction $n$. The glass is moving at constant velocity $v$ in the same direction as the beam and toward the observer in laboratory. What is the speed of light in glass relative to the observer? Take the speed of light in vacuum $c = 1$.

**Bi-NAC**

**1. First response ($y_1$) (Incorrect):**
"Take lab velocity $= v + (1/n)$. So answer $(1/n) + v$."

> **Task-Specific Guidance ($z_l$)**
>
> Apply relativistic velocity addition for velocity $u = 1/n$ inside moving medium with speed $v$. The formula is $(u+v)/(1+uv)$. Simplify to get closed form with $c = 1$.

**2. Improved response ($y_2$) (Correct):**
Inside glass: $u = 1/n$. Using relativistic addition:

$$u' = \frac{u+v}{1+uv} = \frac{1/n+v}{1+v/n} = \frac{1+nv}{n+v}.$$

The final answer is: $\boxed{\frac{1+nv}{n+v}}$.

**GPQA Example 3**

**Prompt:** Find the kinetic energies of the product particles in $\pi^+ \rightarrow \mu^+ + \nu$, where $\pi^+$ is stationary. Rest masses: $m_{\pi^+} = 139.6$ MeV, $m_{\mu^+} = 105.7$ MeV.

**Bi-NAC**

**1. First response ($y_1$) (Incorrect):**
"Split energy equally, $\sim 70$ MeV each."

**Task-Specific Guidance ($z_l$)**

Apply 2-body decay kinematics for stationary parent: Step 1: Write $E_\mu = \frac{m_\pi^2 + m_\mu^2}{2m_\pi}$, and $E_\nu = \frac{m_\pi^2 - m_\mu^2}{2m_\pi}$. Step 2: Subtract the muon's rest mass to get its KE. Step 3: For the massless neutrino, KE equals total energy. Step 4: Plug in values and compute.

**2. Improved response ($y_2$) (Correct):**

// Step 1: Compute muon total energy. $E_\mu = \dfrac{m_\pi^2 + m_\mu^2}{2m_\pi} = \dfrac{(139.6)^2 + (105.7)^2}{2 \times 139.6} \approx 109.8$ MeV.

// Step 2: Subtract rest mass to get muon kinetic energy. $T_\mu = E_\mu - m_\mu = 109.8 - 105.7 \approx 4.1$ MeV.

// Step 3: Compute neutrino energy (massless, so all KE). $E_\nu = \dfrac{m_\pi^2 - m_\mu^2}{2m_\pi} = \dfrac{(139.6)^2 - (105.7)^2}{2 \times 139.6} \approx 29.8$ MeV.

// Step 4: State results. Muon KE $\approx 4.1$ MeV, Neutrino KE $\approx 29.8$ MeV.

The final kinetic energies are: $\boxed{T_\mu = 4.1 \text{ MeV}, \ T_\nu = 29.8 \text{ MeV}}$.

