# OpenReview forum: "RL with Learnable Textual Feedback: A Bilevel Approach"
_ICML.cc/2026/Conference — ICML 2026 regular_

### Official Review · Reviewer_xXwW · 2026-03-10

**Soundness:** 3
**Presentation:** 3
**Significance:** 2
**Originality:** 2
**Overall Recommendation:** 4
**Confidence:** 3

**Summary:**

This paper proposes a framework called Bi-NAC, which aims to address the problem between natural language feedback and the model's reasoning capabilities. Related methods only optimize the actor while neglecting the guiding role of the critic, which results in the feedback that appears correct but is ineffective in fact. The authors believe that the critic and actor should be treated as a bilevel optimization system. The objective of the critic is to generate feedback that improves the performance of the actor. This work proposes Bi-NAC, which models this interaction as a Stackelberg game. Extensive experiments demonstrate Bi-NAC's outstanding performance on multiple reasoning tasks.

**Compliance With Llm Reviewing Policy:**

Affirmed.

**Final Justification:**

The explanations have addressed my concerns. I will keep the positive score.

**Key Questions For Authors:**

1. In Section 3, Advantage Collapse Frequency (74%) and Reward Collapse Frequency (83%) are mentioned, which is presented only as a theoretical overview. Do they only occur at the beginning of the training, or throughout the entire training process? What causes them? Is there any correlation with other hyperparameters, such as batch size or group size?
2. What is the specific value of $\lambda$? Is the method sensitive to variations in $\lambda$? Need some ablation experiments.
3. In Section 4, Q5. The experiments show that there is no significant performance difference between 1M and 2M frameworks. In the single-model framework, how does one model act as both an actor and a critic simultaneously? Please clarify more details.
4. Which model was used to evaluate the FO metric? Does it introduce inherent model preference?

**Limitations:**

yes

**Strengths And Weaknesses:**

Strengths:
1. This paper's motivation is clearly defined and well-supported by data.
2. The authors have conducted a series of experiments across multiple datasets to demonstrate the validity of the method. Particularly, the experiments covered diverse model families (LLaMA & Qwen) and multi-scales (light weight 1B, large weight 7B).
3. The experiments in the ablation study are comprehensive, which provides clear insights into the effectiveness of each individual design component.

Weaknesses:
1. This paper lacks some detailed explanation and ablation studies on the key hyperparameter $\lambda$. Please refer to "Key Questions: Q2", and provide a detailed explanation.
2. Missing the comparison with recent RL methods (such as DAPO, Dr. GRPO, GPG, etc.). Only GRPO is included as a baseline.
3. Lacks a discussion on the method's limitations. It should clarify the scenarios that might cause failure, and identify whether there is sensitivity to the specific data distributions or parameter settings.

---

> ### Author Rebuttal · Authors · 2026-03-31
>
> ### Response to Weaknesses
>
> >**W1 and Q2:**
>
> **Response:** $\lambda$ controls the bilevel penalty strength in Eq. 5: $\lambda=0$ decouples actor-critic, while larger $\lambda$ enforces coupling more aggressively. We provide the $\lambda$ ablation on MATH-500 (LLaMA-3.2-1B):
>
> **Table 6: $\lambda$ Ablation**
> |$\lambda$|MATH-500|
> |--|--:|
> |0.0|41.80|
> |0.1|46.60|
>
> The Performance improves substantially over $\lambda=0$, confirming that bilevel coupling is essential. We are evaluating additional values of $\lambda$, and will report them as soon as we get the results.
>
> >**W2:**
>
> **Response:** We respectfully clarify that our evaluation already includes multiple methods like ArCHer, SCoRe, Critique-GRPO. We additionally compare against Dr. GRPO below. Note: GRPO is evaluated on Llama-3.2-3B; Dr. GRPO on Llama-3.2-FineMath-3B; all others on Llama-3.2-1B:
>
> **Table 7: Additional Baselines Comparison**
> |Method|MATH‑500|
> |--|--|
> |GRPO|26.20%|
> |ArCHer|30.60%|
> |SCoRe|34.80%|
> |Critique‑GRPO|39.60%|
> |Dr. GRPO|38.0%|
> |**Bi‑NAC (Ours)**|**46.60%**|
>
> **Discussion.** Bi‑NAC outperforms all baselines, including larger GRPO and Dr.GRPO. Methods like Dr. GRPO are conceptually orthogonal to our bilevel textual-feedback framework. Specifically, they target single-policy gradient estimation without any critic or textual feedback, whereas Bi-NAC addresses actor-critic alignment through trained textual feedback. We will add a discussion of these in the final version.
>
> > **W3:**
>
> **Response:** We discuss limitations here:
> 1. **Early Training Instability**: The critic may generate ineffective critiques during the zero-reward phase, temporarily stalling both actor and critic learning. Notably, our bilevel formulation mitigates this over time.
> 2. **Verifiable Reward Dependency.** While Bi-NAC is most effective on tasks with deterministic verifiers (math, code), its effectiveness on open-ended tasks with proxy rewards (e.g. creative writing) remains an open question.
> 3. **Test-Time Compute.** Bi-NAC requires two actor passes and one critic pass per query, though this is partially offset by the absence of long chain-of-thought rollouts that substantially inflate inference cost in GRPO-style models.
>
> ### Response to Qns
>
> > **Q1:**
>
> **Response:** The reported Advantage Collapse Frequency (74%) and Reward Collapse Frequency (83%) are **empirical statistics measured throughout GRPO training**, not restricted to the early phase. They were computed on MBPP using LLaMA-3.2-1B with group size G=4.
>
> The root cause is **sparse binary rewards combined with group-based advantage normalization**. When all sampled responses in G are incorrect, each sample receives reward $R_i=0$, so the group mean $\bar{R}=0$ and all normalized advantages collapse to $A_i=0$, eliminating the gradient entirely for that step. While this collapse is worse during the early zero-reward phase, it persists throughout training.
>
> These statistics are also influenced by hyperparameters:
> - **Group size G:** Larger G increases the possibility that at least one response is correct, partially reducing advantage collapse. However, it does not fix per-trajectory reward collapse, since individual failed responses still receive zero reward.
> - **Batch size:** Larger batches reduce gradient variance by increasing the probability that at least one group survives collapse per update step.
>
> Notably, tuning these hyperparameters only masks the symptoms of sparse rewards without solving the underlying sparsity, which is the fundamental issue that motivates our textual feedback approach.
>
> > **Q3:**
>
> **Response:** In the **single-model** setting, the same of set parameters plays both roles via **different input conditioning:**
> - **Actor (Turn 1):** input $x$ → response $y_0$;
> - **Critic:** input $(x, y_0)$ + critique prompt → feedback $z$;
> - **Actor (Turn 2):** input $(x, y_0, z)$ → refined response $y_1$.
>
> The **training** alternates between the actor and critic updates on the same shared parameters, so the model learns both response generation and critique generation within a single backbone.
> - **Actor Update:** After generating the full trajectory $(y_0, z, y_1)$, we clone the current model parameters. Using Eq. 7, we compute the policy gradient to update actor's model weights.
> - **Critic Update:** Using the newly updated weights as the “optimal actor” $(\theta^*)$ for the penalty constraint, we use Eq. 6 to evaluate how effectively the feedback $z$ improved the final task reward. We then apply this to update the model weights.
>
> We also clarify a typo: the “1M” and “2M” entries in Table 3 refer to 1-model and 2-model variants which correspond to a 1B single-model setup and a 2B two-model setup (1B actor + 1B critic), respectively.
>
> > **Q4:**
>
> **Response:** For the FO metric, we utilized GPT-4o as the evaluator model. To mitigate model-preference bias, we use an external judge different from the evaluated models: the feedback is generated by LLaMA backbones, while scoring is done by GPT-4o.

---

> > ### Author Rebuttal · Reviewer_xXwW · 2026-04-02
> >
> > Thank the authors for the comprehensive and thoughtful response. The answers have resolved my questions and clarified the issues I raised. I will retain my weak accept recommendation.

---

> > > ### Author Response · Authors · 2026-04-06
> > >
> > > We sincerely thank the reviewer for their constructive feedback throughout the review process and for confirming that our rebuttal has fully addressed their concerns. We will ensure that the additional experiments and analysis are included in the final manuscript. We appreciate your time and effort in helping us improve the quality of our work.

---

### Official Review · Reviewer_h2Vq · 2026-03-13

**Soundness:** 3
**Presentation:** 3
**Significance:** 3
**Originality:** 3
**Overall Recommendation:** 4
**Confidence:** 4

**Summary:**

This paper studies how to improve LLM reasoning using reinforcement learning with natural language feedback. It proposes Bi-NAC, a bilevel actor–critic framework where a critic generates feedback that is optimized to improve the actor’s final task performance. Experiments on reasoning benchmarks show that the method improves learning efficiency and allows smaller models to outperform larger RL baselines.

**Compliance With Llm Reviewing Policy:**

Affirmed.

**Final Justification:**

Thanks for the response provided by the authors. My concerns have been addressed.

**Key Questions For Authors:**

see weakness

**Strengths And Weaknesses:**

Strengths
- The paper identifies a meaningful limitation in textual actor–critic systems—feedback that appears correct but does not improve the actor’s performance.
- Modeling the system as a bilevel Stackelberg optimization problem is conceptually sound and aligns critic training with actor improvement.
- Experiments across multiple benchmarks show consistent gains in performance, convergence speed, and parameter efficiency.

Weaknesses
- Training two models (actor and critic) may increase compute requirements, but the trade-offs are not thoroughly analyzed.
- While the bilevel framing is interesting, similar hierarchical or multi-agent RL formulations exist; the conceptual novelty relative to prior textual-feedback RL methods could be better clarified.
- The setup resembles a multi-agent learning setting (actor and critic as interacting agents), but the paper does not discuss connections to the multi-agent RL literature or related frameworks.

---

> ### Author Rebuttal · Authors · 2026-03-31
>
> Thank you for your valuable feedback.
>
> ### Response to Weaknesses
>
> > **W1:** Training two ... analyzed.
>
> **Response:** We agree that jointly training an actor and critic increases training-time compute and memory relative to standard single-model GRPO, and we will make this trade-off explicit in the final version.
>
> The key trade-off is that Bi-NAC incurs additional training cost, but converts that cost into improved sample efficiency, parameter efficiency, and final accuracy. For example, Bi-NAC (2B) outperforms GRPO (3B) on both MATH-500 and MBPP, while the single-model Bi-NAC (1B) remains close to the two-model variant and reduces deployment overhead.
>
> **Table 5:Compute-performance trade-off across GRPO and Bi-NAC variants.**
> | Method | Architecture | Total Parameters | Training Time (hrs) |Inference Time (s) |MATH-500 Accuracy | MBPP  Accuracy |
> |---|---|---:|---:|---:|---:|---:|
> | GRPO | Single actor | 3B |18.5|8.3| 41.43 | 61.55 |
> | Bi-NAC | 1B actor + 1B critic | 2B |16.5|9.7| 46.56 | 66.73 |
> | Bi-NAC | Single model | 1B |16.5|9.7| 46.84 | 65.24 |
>
> As shown in Table 5, Bi-NAC (1B actor + 1B critic, 2B total) achieves **46.56%** on MATH-500 and **66.73%** on MBPP while training in **16.5** hours, as compared to GRPO's **41.43%** and **61.55%** with a 3B model trained in **18.5** hours. This means Bi-NAC achieves higher accuracy with fewer total parameters and less training time, demonstrating that our bilevel formulation improves sample efficiency.  Further, using only 1B parameters, the single-model Bi-NAC variant matches the two-model variant on both benchmarks, while reducing inference overhead to near parity with standard GRPO.
>
> We also clarify a typo in the manuscript: the “1M” and “2M” entries in Table 3 refer to 1-model and 2-model variants, not million-scale models. Concretely, they correspond to a 1B single-model setup and a 2B two-model setup (1B actor + 1B critic), respectively.
>
> > **W2:** While ... clarified.
>
> **Response:**  Thank you for this helpful comment. Our conceptual novelty relative to prior textual-feedback RL methods is that **Bi-NAC does not treat feedback generation as fixed, auxiliary, or decoupled from the actor's downstream learning signal.** Instead, we formulate feedback generation itself as a **bilevel optimization problem**, where the critic is explicitly trained to produce feedback that improves the actor’s downstream verifiable reward.
>
> - This is the key difference from prior methods such as SCoRe [3] and Critique-GRPO [4], where feedback is generated either from fixed prompts or from an auxiliary critic that is not optimized to improve the actor's downstream performance. In Bi-NAC, the critic's training signal is the actor's verifiable reward on the refined response, which makes the feedback aligned to the actor.
> - Relative to broader hierarchical RL formulations that use natural language subgoals [5], our setting is also distinct in that the higher-level policy produces textual critiques and is optimized for the lower-level actor’s refined task success in a Stackelberg-style bilevel setup. We will clarify this distinction more explicitly in the final version.
>
> [3] Training Language Models to Self-Correct via Reinforcement Learning (SCoRe), Kumar et al., 2024.
>
> [4] Critique-GRPO: Advancing LLM Reasoning with Natural Language and Numerical Feedback, Zhang et al., 2025.
>
> [5] Hierarchical Reinforcement Learning with Natural Language Subgoals, Ahuja et al., 2023.
>
> > **W3:** The setup ... frameworks.
>
> **Response:** We agree that Bi-NAC has a natural connection to the multi-agent RL literature, since it involves two interacting policies with coupled objectives. We will add a short discussion in the final version to better position our method relative to MARL and related hierarchical/game-theoretic formulations.
>
> At the same time, we will clarify that Bi-NAC is not a standard MARL setup: the interaction is **sequential and asymmetric**, with the critic producing **textual feedback** and the actor refining its response conditioned on that feedback. We will also cite the relevant MARL references [6,7] in the revised manuscript.
>
> [6] Lowe et al., Multi-Agent Actor-Critic for Mixed Cooperative-Competitive Environments, NeurIPS 2017.
> [7] Foerster et al., Counterfactual Multi-Agent Policy Gradients, AAAI 2018.

---

> > ### Author Rebuttal · Reviewer_h2Vq · 2026-04-03
> >
> > Thank the authors for their comprehensive and thoughtful response. I will keep my score.

---

> > > ### Author Response · Authors · 2026-04-06
> > >
> > > We sincerely thank the reviewer for their constructive feedback throughout the review process and for confirming that our rebuttal has fully addressed their concerns. We will ensure that the additional experiments and analysis are included in the final manuscript. We appreciate your time and effort in helping us improve the quality of our work.

---

### Official Review · Reviewer_5N8k · 2026-03-13

**Soundness:** 3
**Presentation:** 3
**Significance:** 3
**Originality:** 3
**Overall Recommendation:** 5
**Confidence:** 4

**Summary:**

The paper proposes Bi-NAC, which divides the system into two layers and unifies feedback generation and policy improvement into a two-layer optimization framework. The authors conducted experiments on three inference tasks: MATH-500, MBPP, and GPQA.

**Compliance With Llm Reviewing Policy:**

Affirmed.

**Final Justification:**

This paper addresses a meaningful and worthwhile research problem. The experimental analysis is rigorous, and the work provides solid theoretical justification. During the rebuttal phase, the authors offered thorough and convincing responses. After considering the questions from other reviewers and the authors’ replies, I believe the concerns have been adequately addressed and recommend 5: Accept !

**Key Questions For Authors:**

1. 2B Bi-NAC surpasses 3B GRPO, and 6B Bi-NAC surpasses 7B GRPO. Are the rollout count, update steps, and sampling costs comparable? How was this comparison made?
2. The paper emphasizes that existing methods treating the critic as a fixed/auxiliary module leads to a mismatch problem where feedback is correct but invalid. Is this problem prevalent in different post-training algorithms? How can this phenomenon be proven?
3. What are the innovations and improvements in final performance of Bi-NAC compared to SCoRe, Critique-GRPO, and ArCHer: the bilevel target itself, the critic training method, or the joint update mechanism of actor/critic?
4. Has the dependency of $\theta^{*}(\phi)$ on $\phi$ been ignored? Is this theoretically valid?

I will adjust my final score during the rebuttal period based on other reviewers' suggestions and authors' responses.

**Limitations:**

yes, the authors adequately discuss the limitations and potential negative societal impact of their work.

**Strengths And Weaknesses:**

Strengths: The authors reformulate reasoning reinforcement learning with natural language feedback as a two-layer actor-critic problem and propose Bi-NAC, which transforms feedback from merely auxiliary information into a learnable signal that can truly improve the final reasoning performance. The authors have conducted extensive experimental verification.

Weaknesses: Some theoretical analyses and experimental results in this paper may be incomplete; please refer to Key Questions For Authors for details.

---

> ### Author Rebuttal · Authors · 2026-03-31
>
> ### Response to Questions
>
> > **Q1:**
>
> **Response:** To verify fair comparison, we report training budgets below. Bi-NAC uses fewer tokens per sample (5,120 vs. 16,384) due to shorter critique-based rollouts, partially offsetting the cost of training two models. The total compute is comparable or lower despite the two-model setup.
>
> |Method|Model Size|Rollouts|Update Steps|Tokens/Sample|
> |--|--|--|--|--|
> |GRPO|3B|32|1,600|16,384|
> |**Bi-NAC**|**2B (2×1B)**|**128**|**1,600**|**5,120**|
> |GRPO|7B|128|1,600|16,384|3,355,443,200|
> |**Bi-NAC**|**6B (2×3B)**|**128**|**1,600**|**5,120**|
>
> *Bi-NAC 6B uses 3.2× fewer tokens per sample and 3.2× less total compute than GRPO 7B, confirming the gains are not due to an inflated compute budget. Thus, Bi-NAC requires significantly less compute to surpass GRPO's final accuracy.*
>
> > **Q2:**
>
> **Response:** Yes, this mismatch is prevalent in any method where feedback is decoupled from actor adaptation. The example below (Prompt: $2^8 = 4^x$, find $x$) illustrates that the usefulness of feedback is not determined only by its correctness, but by whether conditioning on it leads to higher downstream performance.
>
> |Setting|Feedback|Correct in isolation?|Refined answer|
> |--|--|--|--|
> |Fixed feedback (SCoRe)|“The solution is incorrect. Please try again.”|Yes|$x=8$ (Wrong)|
> |Auxiliary critic (Critique-GRPO)|“Base conversion is correct. From $3x = x + 6$, subtracting gives $x = 3$.”|Yes|$x=3$ (Wrong)|
> |Bi-NAC aligned critic (ours)|"Rewrite 4 as $2^2$, Apply the power-of-a-power rule to obtain $2^8 = 2^{2x}$. Solve for x."|Yes|$x=4$ (Correct)|
>
> In above example, in the case of fixed or auxiliary model generated feedback, although the feedback may be correct, it may be unable to generate correct responses. In contrast, Bi-NAC predicts actor-aligned feedback, which leads to improved downstream performance.
>
> **Empirical Proof (Table 2).** We quantify this via **Feedback Optimality** (FO, LLM-judged 1–5) and **Feedback Compatibility** ($\Delta_{i\to c}$, fraction flipped incorrect→correct). The **Hier-FT** baseline achieves higher FO (**3.0**) than **SCoRe** (**2.1**) but lower compatibility (**16.0%** vs. **17.8%**), directly proving that feedback correctness does not imply feedback utility. Bi-NAC resolves this by coupling critic learning with actor reward, achieving FO=**4.2**, compatibility=**21.2%**, and accuracy=**46.6%**.
>
> > **Q3:**
>
> **Response:** Our main innovations are **bilevel target** and **critic training** method. The **joint actor-critic update** is the optimization mechanism that enables them in practice. The comparison table below (Llama-3.2-1B) compares Bi-NAC with prior methods:
>
> |Method|Bilevel Target|Critic Training|Joint Actor/Critic Update|MATH-500 Accuracy|
> |--|--|--|--|--|
> |ArCHer|No|No (RL actor only)|No|30.6%|
> |SCoRe|No|No (Fixed feedback)|No|34.8%|
> |Critique-GRPO|No|No (Auxiliary feedback)|No|39.6%|
> |**Bi-NAC**|**Yes**|**Yes**|**Yes**|**46.6%**|
>
> The main innovations of Bi-NAC are:
> (1) a bilevel target that trains the critic for actor improvement,
> (2) a learned critic instead of fixed or auxiliary feedback,
> (3) a joint update mechanism that keeps actor and critic aligned during training.
>
> > **Q4:**
>
> **Response:** We do not ignore the dependency of $\theta^\*(\phi)$ on $\phi$. It is ***one of our key contributions***: the upper-level objective in Eq. (3) is defined through the actor solution $\theta^\*(\phi)$, so the critic is optimized while accounting for how the actor adapts to its feedback.
>
> **Key challenge:** For large LLMs (e.g., LLaMA-3.2-1B/8B), handling exact dependency of $\theta^*(\phi)$ on $\phi$ through the exact implicit gradient is computationally intractable [2], as that involves computing the inverse Hessian matrix of the lower-level objective.
>
> To address this without ignoring the dependency, we rely on recent advances in scalable value-function-based bilevel optimization [1]. Instead of directly differentiating to find $\nabla_\phi \theta^\*(\phi)$, we reformulate the Stackelberg game into a constrained optimization problem (Eq. 4) and derive its Lagrangian (Eq. 5) (without any loss or approximation from the original bilevel formulation). This reformulation **does not discard the dependence** on $\theta^\*(\phi)$, and rewrites the bilevel problem in an equivalent constrained form.
>
> We do not compute the exact implicit hypergradient through $\theta^*(\phi)$, thus avoiding the intractable second-order bilevel operations such as inverse-Hessian terms. Instead, **inspired by recent scalable bilevel optimization methods**, we optimize the resulting first-order Lagrangian objective, yielding the critic & actor updates in Eqs. (6) and (7).
>
> Thus, the dependence is preserved through an equivalent constrained reformulation, while the optimization avoids exact second-order implicit differentiation for scalability.
>
> [1] Bome! Bilevel optimization made easy, Liu et al., 2022.
>
> [2] Approximation Methods for Bilevel Programming, Ghadimi and Wang, 2018.

---

> > ### Author Rebuttal · Reviewer_5N8k · 2026-04-03
> >
> > My confusion has been initially resolved, I would be very willing to raise my score.

---

> > > ### Author Response · Authors · 2026-04-06
> > >
> > > We sincerely thank the reviewer for their constructive feedback throughout the review process and for confirming that our rebuttal has fully addressed their concerns. We will ensure that the additional experiments and analysis are included in the final manuscript. We appreciate your time and effort in helping us improve the quality of our work.

---

### Official Review · Reviewer_wY9Z · 2026-03-13

**Soundness:** 3
**Presentation:** 3
**Significance:** 3
**Originality:** 3
**Overall Recommendation:** 4
**Confidence:** 4

**Summary:**

This paper introduces Bi-NAC, a bilevel optimization framework for training natural language actor-critic systems in LLM reasoning. The core insight is that existing approaches suffer from feedback misalignment: a critic that generates textually correct feedback may not actually help the actor improve, because the two are trained independently. Bi-NAC formalizes this as a Stackelberg game where the critic (upper level) optimizes feedback to maximize the actor's (lower level) downstream task reward, with gradients derived from a Lagrangian relaxation of the bilevel program. Using a fixed 2-turn interaction (initial response, critic feedback, refined response), experiments on MATH-500, MBPP, and GPQA with LLaMA and Qwen models (1B–8B) show that Bi-NAC consistently outperforms GRPO, SCoRe, and other baselines, with a 2B Bi-NAC surpassing 3B GRPO, demonstrating strong parameter and sample efficiency.

**Compliance With Llm Reviewing Policy:**

Affirmed.

**Final Justification:**

This paper extends GRPO into an actor-critic paradigm with a bilevel formulation, demonstrating the viability and benefit of jointly training a critic alongside the actor for reasoning tasks. This contributes to making actor-critic methods more practical in the LLM reasoning setting. After rebuttal, most of my concerns have been well addressed.

**Key Questions For Authors:**

See weaknesses

**Limitations:**

Yes

**Strengths And Weaknesses:**

## Strengths

- This paper extends GRPO into an actor-critic paradigm with a bilevel formulation, demonstrating the viability and benefit of jointly training a critic alongside the actor for reasoning tasks. This contributes to making actor-critic methods more practical in the LLM reasoning setting.

- The paper is well written and easy to follow.

- The paper addresses the important and underexplored direction of optimizing textual feedback itself rather than treating it as a fixed input, which is a meaningful research contribution beyond standard RLVR methods.

- The proposed feedback optimality and feedback compatibility metrics provide a quantitative framework for evaluating feedback quality in actor-critic systems, which could be useful for future work in this direction.

## Weaknesses

- The evaluation benchmarks (MATH-500, MBPP, GPQA) are relatively easy for current frontier models (e.g., Qwen-2.5, Qwen-3). It is unclear whether the proposed method would show similar gains on harder benchmarks such as AIME24, where the base models struggle more significantly and the feedback alignment problem may manifest differently.

- The framework is limited to a fixed 2-turn interaction (initial response + one refinement). The paper does not explore whether extending to more refinement turns at training or inference time could yield further improvements. Given that the bilevel formulation already learns an aligned critic, it would be natural to investigate whether iterative multi-turn refinement could compound the gains, and the absence of this analysis is a notable gap.

---

> ### Author Rebuttal · Authors · 2026-03-31
>
> Thank you for your valuable feedback.
>
> ### Response to Weaknesses
> > **W1:** The evaluation benchmarks (MATH-500, MBPP, GPQA) are relatively easy for current frontier models (e.g., Qwen-2.5, Qwen-3). It is unclear whether the proposed method would show similar gains on harder benchmarks such as AIME24, where the base models struggle more significantly and the feedback alignment problem may manifest differently.
>
> **Response to W1:** We address this by adding evaluations on AIME24, AIME25, and BeyondAIME. in Table 1, using Qwen3-1.7B (non-thinking mode) as the base model. Results are reported at Turn 1 (no refinement) and Turn 2 (one critic-guided refinement). This suggests the critic-guided refinement mechanism remains effective even on substantially harder benchmarks.
>
> **Table 1: AIME Benchmark Results (Qwen3-1.7B, non-thinking)**
>
> | Benchmark | Method | Turn 1 | Turn 2 | Δ |
> |---|---|---|---|---|
> | AIME 2024 | Untrained baseline | 6.7% | 6.7% | +0.0% |
> | | **Bi-NAC (ours)** | 6.7% | **13.3%** | **+6.6%** |
> | AIME 2025 | Untrained baseline | 6.7% | 6.7% | +0.0% |
> | | **Bi-NAC (ours)** | 6.7% | **10.0%** | **+3.3%** |
> | BeyondAIME | Untrained baseline | 2.0% | 2.0% | +0.0% |
> | | **Bi-NAC (ours)** | 2.0% | **4.0%** | **+2.0%** |
>
> *Turn 1 = no refinement (identical across methods by construction). Turn 2 = one critic-guided refinement.*
>
> ---
>
> > **W2:** The framework is limited to a fixed 2-turn interaction (initial response + one refinement). The paper does not explore whether extending to more refinement turns at training or inference time could yield further improvements. Given that the bilevel formulation already learns an aligned critic, it would be natural to investigate whether iterative multi-turn refinement could compound the gains, and the absence of this analysis is a notable gap.
>
> **Response to W2:** We thank the reviewer for this helpful suggestion. In response, we provide multi-turn inference results on MATH-500 using Qwen3-1.7B (non-thinking mode) in Table 2.
>
> - The results show that the largest gain occurs from Turn 1 to Turn 2 ($\sim$ 7 - 8%), with a modest additional improvement at Turn 3 (~ 1–2%), after which performance plateaus. This saturation effect motivates our choice of the 2-turn training setup as the most compute-efficient configuration.
> - Importantly, Bi-NAC's trained critic enables the model to realize these gains earlier and more reliably than the untrained baseline. We plan to extend the training-time setup to multi-turn in the final version.
>
> **Table 2: Multi-Turn Accuracy on MATH-500 (Qwen3-1.7B, non-thinking)**
>
> | Method | Turn 1 | Turn 2 | Turn 3 | Turn 4 | Turn 5 |
> |---|---|---|---|---|---|
> | Untrained multi-turn baseline | 62.4% | 65.6% | 62.2% | 64.7% | 63.3% |
> | **Bi-NAC (ours)** | 62.4% | **70.2%** | **71.6%** | 71.0% | 70.8% |
>
> We hope these responses clarify the reviewer's concerns. Please let us know if you need further clarifications.

---

> > ### Author Rebuttal · Reviewer_wY9Z · 2026-04-03
> >
> > Thanks for the detailed rebuttal. My concerns have been addressed. And I will rasie my rating.

---

> > > ### Author Response · Authors · 2026-04-06
> > >
> > > We sincerely thank the reviewer for their constructive feedback throughout the review process and for confirming that our rebuttal has fully addressed their concerns. We will ensure that the additional experiments and analysis are included in the final manuscript. We appreciate your time and effort in helping us improve the quality of our work.

---

### Decision · Program_Chairs · 2026-04-30

**Decision:**

Accept (regular)

**Comment:**

This paper proposes a bilevel optimization algorithm that jointly trains a critic and actor for LLM reasoning, where the critic generates natural language feedback optimized to improve the actor's task reward. The key insight is that training the feedback abilitiy of the critic improves the actor's multi-turn reasonign capability. All reviewers agree the bilevel formulation is novel and well-motivated, experiments span multiple benchmarks and model families, and the paper is clearly written.

The authors were responsive during rebuttal, but several concerns are partially resolved:

- Hard benchmarks (wY9Z): AIME results were added, but only on Qwen3-1.7B non-thinking mode with very low base accuracy (2–6.7%). The absolute improvements are small (e.g., 2%→4% on BeyondAIME), making it hard to draw strong conclusions. Crucially, no experiments test frontier-scale or thinking-enabled models, where the self-correction gap Bi-NAC targets may already be smaller.
- Compute comparison (5N8k, h2Vq): The trade-off table is helpful but the compute comparison looks more like a training configuration comparison instead of the actual compute comparison. The claim of "less compute" deserves more careful normalization.
- λ ablation (xXwW): Only λ=0 and λ=0.1 were tested. The authors acknowledged additional values are pending. This is insufficient to characterize sensitivity to the most important hyperparameter in the framework.
- Bilevel dependency (5N8k Q4): The explanation is reasonable, i.e. using a constrained Lagrangian reformulation to avoid intractable second-order terms, but whether this approximation introduces meaningful error is not empirically validated.

Overall, the paper addresses a real problem with a principled framework and consistent empirical gains at small scale. I recommend weak accept, given the comprehensive responses during the rebuttal period.

Suggested revisions:
- Complete the λ ablation with a sufficient range of values and report sensitivity analysis.
- Normalize compute comparisons (e.g., total FLOPs or GPU-hours) rather than relying on training configuration (max generation length, rollout number)
- Add the promised MARL discussion and limitations section.
- Explicitly discuss the method's applicability to larger and thinking-enabled models as a limitation. If possible, include at least one experiment at the 7B+ scale on a hard benchmark to strengthen the claims.